

# Aerosol optical properties and trace gas emissions by PAX and OP-FTIR for laboratory-simulated western US wildfires during FIREX

Vanessa Selimovic[1], Robert J. Yokelson[1], Carsten Warneke[2], James M. Roberts[2], Joost de Gouw[3], James Reardon[4], David W. T. Griffith[5]

[1]Department of Chemistry, University of Montana, Missoula, 59812, USA
[2]Chemical Sciences Division, Earth System Research Laboratory, National Oceanic and Atmospheric Administration, Boulder, CO 80305, USA
[3]Cooperative Institute for Research in Environmental Sciences, University of Colorado, Boulder, CO 80309, USA
[4]USDA Forest Service, Rocky Mountain Research Station, Fire Sciences Laboratory, Missoula, MT, USA
[5]Department of Chemistry, University of Wollongong, Wollongong, New South Wales, 2522, Australia

*Correspondence to:* R. J. Yokelson (bob.yokelson@umontana.edu)

**Abstract.** Western wildfires have a major impact on air quality in the US. In the fall of 2016, 107 test fires were burned in the large-scale combustion facility at the US Forest Service Missoula Fire Sciences Laboratory as part of the Fire Influence on Regional and Global Environments Experiment (FIREX). Canopy, litter, duff, dead wood, and other fuel components were burned in combinations that represented realistic fuel complexes for several important western US coniferous and chaparral ecosystems including Ponderosa Pine, Douglas Fir, Engelmann Spruce, Lodgepole Pine, Subalpine Fire, Chamise, and Manzanita In addition, dung, Indonesian peat, and individual coniferous ecosystem fuel components were burned stand-alone to investigate the effects of individual components (e.g. "duff") and fuel chemistry on emissions. The smoke emissions were characterized by a large suite of state-of-the-art instruments. In this study we report emission factor (EF, g compound emitted per kg fuel burned) measurements in fresh smoke of a diverse suite of critically-important trace gases measured by open-path Fourier transform infrared spectroscopy (OP-FTIR). We also report aerosol optical properties (absorption EF, single scattering albedo (SSA), and Ångström absorption exponent (AAE)) as well as black carbon (BC) EF measured by photoacoustic extinctiometers (PAX) at 870 and 401 nm. The average trace gas emissions were similar across the coniferous ecosystems tested and most of the variability observed in emissions could be attributed to differences in the consumption of components such as duff and litter, rather than the dominant tree species. Chaparral fuels produced lower EF than mixed coniferous fuels for most trace gases except for $NO_x$ and acetylene. A careful comparison with available field measurements of wildfires confirms that several methods can be used to extract data representative of real wildfires from the FIREX lab fire data. This is especially valuable for species not yet measured in the field. For instance, the OP-FTIR data alone show that ammonia (1.65 g kg$^{-1}$), acetic acid (2.44 g kg$^{-1}$), nitrous acid (HONO, 0.61 g kg$^{-1}$) and other trace gases such as glycolaldehyde and formic acid are significant emissions not previously measured for US wildfires. The PAX measurements show that the ratio of brown carbon (BrC) absorption to BC absorption is strongly dependent on modified combustion efficiency (MCE) and that BrC absorption is most dominant for combustion of duff (AAE 7.13) and rotten wood (AAE 4.60): fuels that are consumed in greater amounts during wildfires than prescribed fires. Coupling our lab data with field data suggests that fresh wildfire smoke typically has an EF for BC near 0.1 g kg$^{-1}$), an SSA of ~0.91 and an AAE of ~3.50, with the latter implying that about 86% of the aerosol absorption at 401 nm is due to BrC.

## 1 Introduction

Biomass burning (BB) is a year-round global phenomenon that plays an important role in the budget of many species in atmospheric chemistry. BB can be natural (e.g. wildfire) or anthropogenic (e.g. cooking and agricultural fires) (Crutzen and Andreae, 1990). BB is the largest global source of fine primary organic aerosol (OA), black carbon (BC), and brown carbon



(BrC) (Bond et al., 2004, 2013; Akagi et al., 2011); and the second largest source of $CO_2$, total greenhouse gases, and non-methane organic gases (NMOG) (Yokelson et. al, 2008; 2009), which are precursors for the formation of ozone and OA. About 80% of BB occurs in the tropics, but even the small fraction of total BB in the western US is responsible for significant US air quality impacts (Park et al., 2007; Liu et al., 2017). Record high temperatures, drought, and fire-control practices over the last

century have culminated into a situation in which we can expect more frequent fires and fires of a larger size and intensity in the Western US and Canada (Yue et al., 2015; Westerling et al., 2006). While wildfires are understood to be a natural part of many ecosystems, modern day practices have led to an accumulation of fuels and a breakdown in the natural ecology of forests, leading to a disequilibrium; notable in the form of increased fire risk and fire behavior that is more difficult to control (Stevens et al., 2014). Prescribing fires and reducing aggressive fire suppression techniques are options to remedy the situation, but factors not

related to the direct risk of fire, such as atmospheric impacts of smoke on air quality, climate, and health are still a concern. Despite these important atmospheric chemistry issues, much of the emissions from BB remain either understudied or completely unstudied. To date, most of the research on the emissions and evolution of smoke from US fires has targeted prescribed fires (Burling et al., 2011; Akagi et al., 2013; Yokelson et al., 2013; May et al., 2014; Müller et al., 2016). However, wildfires burn a different mix of fuels in a different season that has more intense photochemistry and different smoke dispersion scenarios, and

they typically consume more fuel per unit area than prescribed fires and can have different emission factors (EF, g compound emitted/kg fuel burned ) (Campbell et al., 2007; Yokelson et al., 2013; Urbanski, 2013). For instance, Liu et al. (2017) found that wildfires had an average EF for $PM_1$ (particulate matter with an aerodynamic diameter less than 1 micron) more than two times that of prescribed fires and that wildfire $PM_1$ was more OA dominated. Despite the large BB emissions of greenhouse gases and BC, it has been assumed that BB OA contributes to negative radiative forcing by BB overall. However, the overall BB forcing

could be positive if the emitted weakly-absorbing OA known as brown carbon (BrC) is sufficiently absorbing and long lived (Feng et al., 2013; Jacobsen, 2014; Saleh et al., 2014; Forrister et al., 2015). This could generate a positive feedback with the expected increase in BB due to a warming climate (Feng et al., 2013; Doerr and Santin, 2016; Bowman et al., 2017). Thus, comprehensive understanding of wildfire contributions to air quality and climate requires further evaluation.

The Fire Influence on Regional and Global Environments Experiment (FIREX) (https://www.esrl.noaa.gov/csd/projects/firex/)
multi-year campaign led by the National Oceanic and Atmospheric Administration (NOAA) aims to answer research questions and critical unknowns about BB that can be addressed with existing or new technologies, laboratory and field studies, and interpretive efforts in order to understand and predict the impact of North American fires on the atmosphere and ultimately support land management. The first phase of this multi-year campaign took place at the US Forest Service Fire Sciences Laboratory (FSL) in Missoula, Montana in the fall of 2016. We deployed a comprehensive suite of standard instrumentation as

well as newer measurement techniques and analysis methods to better assess BB emissions. Each approach has its strengths and weaknesses and many uncertainties are difficult to quantify based on data from a single instrument. Thus, combining results from many techniques to develop a larger data set is critical to achieving the fullest understanding of the capabilities of each method and to better comprehend the full diversity of the emissions and their impacts. Lab fires provide the most cost-effective opportunity to deploy a large suite of instruments and test new instruments under conditions with realistic concentration ranges,

and sample matrix effects such as interferences. Fuel composition and the ambient conditions under which the fuel burned are better known in a laboratory. Additionally, only in a laboratory setting can essentially all of the smoke from a fire be sampled, so that sampling errors are minimized. However, accurate lab-based EF are most valuable when they result from burning realistically re-created fuels from complex flammable ecosystems that produce emissions representative of field fires (Yokelson et al., 2013). Thus, we simulated the fuel and combustion conditions of real wildfires to the extent possible in hopes of obtaining

the most relevant emissions measurements.



As part of the first (laboratory) phase of FIREX we deployed an open-path Fourier transform infrared spectrometer (OP-FTIR) and two photoacoustic extinctiometers (PAX) operating at 401 nm and 870 nm. In this paper, based on these instruments we report EFs for 22 trace gases, BC, and absorption ($B_{abs}$) at two wavelengths for 31 stack burns (stack burns are defined later), along with the single scattering albedos (SSA) and the Ångström absorption exponents (AAE). We also report the trace gas and

BC EFs, along with EF Babs and SSA at just 870 nm for another 44 stack fires. After the first 31 fires, our 401 nm PAX was moved and sampled from a barrel as part of an intercomparison, while the 870 nm stayed sampling the remaining stack fires. After all the stack fires were done, the 870 nm PAX moved to participate in an additional intercomparison of aerosol optical property measurement techniques carried out in BB aerosol. The intercomparison results will be reported elsewhere (Li et al., in preparation; Wagner et al., in preparation, Manfred et al., 2017). In this paper, we examine how well we succeeded in our goal of

obtaining emissions data representative of real wildfires, how the fuels influenced the emissions, and we highlight some of the important species that we measured during FIREX that are still unmeasured in real wildfires.

## 2 Experimental details

### 2.1 U.S. Forest Service Fire Science Laboratory (FSL)

The FSL has a large indoor combustion room described in more detail elsewhere (Christian et al., 2003; Burling et al., 2010). Briefly, the room is 12.5 m × 12.5 m × 22 m high with a 1.6 m diameter exhaust stack and a 3.6 m inverted funnel opening approximately 2 m above a continuously weighed fuel bed. The room can be pressurized to create a large constant flow that dilutes and completely entrains the fire emissions while venting through the stack.  A sampling platform that can accommodate up to 1820 kg and sampling ports surround the stack 17 m above the fuel bed. Other instrumentation can be placed in adjacent

rooms with sample lines pulling from ports at the sampling platform. Previous studies concluded that the temperature and mixing ratios are consistent across the width of the stack at the height of the platform, confirming well-mixed emissions that can be monitored by a number of different sample lines throughout the fire (Christian et al., 2004).

Our simulated fires used two configurations. In the first configuration, termed "stack burns," fires were ignited below the stack and they burned for a few minutes to half an hour. As the fire evolved, the emissions, partially diluted and cooled by outside air,

traveled up through the stack at a constant flow rate (~3.3. m s$^{-1}$). At the platform height, the well mixed emissions were near ambient temperature, about 5 s old and monitored by a large range of instruments in real time.

In the second scenario, referred to as "room burns," most of the instruments were relocated to rooms adjacent to the combustion chamber and used sample lines that extended well within the combustion room. The stack was raised, the combustion room was sealed and the fuels burned for several minutes. After about 15-20 minutes, the smoke from the whole fires was well mixed

vertically in the combustion room and was monitored under approximately steady-state, low light conditions for up to several hours; though some infiltration and losses of gases and particles for some species occurred (Stockwell et al., 2014). Despite the losses, the configuration is useful for measurements requiring longer times. The OP-FTIR remained on the sampling platform during room burns, which helped to document the initial rise of flaming emissions and verified the overall mixing processes. Temperature and relative humidity in the combustion room were recorded for all fires and both stack and room burns were

videotaped and stored in the NOAA archive.



## 2.2 Fuels

A team of experts collected fuels that represent fire-prone Western U.S ecosystems primarily from the Clearwater Game Range (http://fwp.mt.gov/fishAndWildlife/wma/siteDetail.html?id=39754079) and the Lubrecht Experimental Forest (https://www.cfc.umt.edu/lubrecht/), which are managed by the State and University of Montana, respectively. Chaparral fuels and fuels for the Fire and Smoke Model Evaluation Experiment (FASMEE, https://www.fasmee.net/) were sampled by forest fire experts at locations in California and Utah, respectively, and shipped overnight to the FSL. A few fuels representative of prescribed fires were sampled by foresters at SE US military bases and burned for comparison purposes and for the FASMEE project. Sagebrush and juniper were sampled locally. Indonesian peat, poplar shavings, and dung were sampled and burned because of their global importance and/or to investigate the impact of fuel chemistry (e.g. N content) on emissions. Fuel "components" for the forest ecosystems included duff, litter, dead and down woody debris in different size classes, herbaceous, shrub, and canopy fuels, as well as rotten logs from two of the above ecosystems (Ponderosa Pine and Douglas Fir). These fuel components were burned both on their own, standalone, and in realistic three-dimensional mixtures to mimic the different fuel complexes for various ecosystems. The first order fire effects model (FOFEM) (Reinhardt et al., 1997) was used to calculate the relative amount of each component that typically burns in coniferous ecosystems, while pure components were burned to probe how they affected the total emissions. The coniferous ecosystems modeled and burned included Ponderosa pine *(Pinus ponderosa)*, Lodgepole pine *(Pinus contorta)*, Engelmann spruce *(Picea engelmanii)*, Douglas fir *(Pseudotsuga menziesii)*, and Sub-alpine fir *(Abies lasiocarpa)*. Chaparral was represented by Manzanita *(Arctostaphylos)* and Chamise *(Adenostoma fasciculatum)*. A full description of the fuels for each fire, including collection location; C, H, N, S, and Cl content; dry weight of each component; ignition time; etc. is included in Table S1. Moisture content, ash data, and other details for fuels and fire duration were also recorded, and are available in the NOAA archive or from the corresponding author.

## 2.3 Instrument details

Extensive Instrumentation that monitored both the gas-phase and particle phase-emissions from BB was deployed during the FIREX lab study. A table of all the instruments can be found at this URL (https://www.esrl.noaa.gov/csd/projects/firex/firelab/instruments.html) and more details will be in a companion paper by Roberts et al (in preparation). We reiterate that for the first 31 stack fires the two PAXs were the only instruments measuring aerosol optical properties on the platform and only the 870 nm PAX measured optical properties on the sampling platform for the next 44 fires, which accounts for all the stack burns. The 401 nm PAX was deployed with a BC intercomparison that measured subsamples of smoke in a mixing barrel for fires 32-107. The 870 nm PAX was deployed with a large group of aerosol instruments that characterized aerosol subsamples from the room burns (fires 76-107). Other aerosol measurements on the platform during the stack burns included filter sampling with off-line analysis of non-methane organic compounds and AMS characterization of diluted smoke. Here we present the PAX (and FTIR) measurements on the platform and the other results will be described elsewhere.

### 2.3.1 Open-path Fourier transform spectrometer (OP-FTIR)

The OP-FTIR consisted of a Bruker Matrix-M IR Cube spectrometer with a mercury-cadmium-telluride (MCT) liquid nitrogen cooled detector interfaced with a 1.6 m base open-path White cell. The optical path length was 58 m and IR spectra were collected at a resolution of 0.67 cm$^{-1}$ from 600-4000 cm$^{-1}$. During stack burns, the OP-FTIR was positioned on the sampling platform so that the open path fully spanned the width of the stack. This allowed continuous direct measurements across the





rising emissions. A pressure transducer and two temperature sensors were located directly adjacent to the White cell optical path
and used for spectrum fitting and to calculate mixing ratios from the IR spectra. For stack burns the time resolution was
approximately 1.37 seconds and the duty cycle was >95%. For the room burns, where concentrations changed more slowly, we
increased the sensitivity by co-adding scans (time resolution to approximately 5.48 seconds) and moved the OP-FTIR to the edge
of the sampling platform closest to the fires.

Mixing ratios were determined for carbon dioxide ($CO_2$), carbon monoxide (CO), methane ($CH_4$), acetylene ($C_2H_2$), ethylene
($C_2H_4$), propylene ($C_3H_6$), 1,3-butadiene ($C_4H_6$), formaldehyde (HCHO), formic acid (HCOOH), methanol ($CH_3OH$), acetic acid
($CH_3COOH$), glycolaldehyde ($C_2H_4O_2$), furan ($C_4H_4O$), furaldehyde ($C_5H_4O$), phenol ($C_6H_6O$), hydroxyacetone ($C_3H_6O_2$), water
($H_2O$), nitric oxide (NO), nitrogen dioxide ($NO_2$), nitrous acid (HONO), ammonia ($NH_3$), hydrogen cyanide (HCN), hydrogen
chloride (HCl), and sulfur dioxide ($SO_2$). Mixing ratios are based on retrievals utilizing multi-component fits to specific sections

of mid-IR transmission spectra with a synthetic calibration non-linear least-squares method (Griffith, 1996, Yokelson et al.,
2007) applying both the HITRAN spectral database and reference spectra recorded at Pacific Northwest National Laboratory
(PNNL) (Rothman et al., 2009; Sharpe et al., 2004; Johnson et al., 2013; Johnson et al., 2010). The above species were always or
often enhanced in the smoke well above the real-time detection limits, but some species such as 1,3-butadiene, furaldehyde,
phenol, and HCl were frequently not enhanced more than 2-3 times the real-time detection limit and are not reported in those

cases. The uncertainties in the individual mixing ratios varied by spectrum and molecule and were influenced by uncertainty in
the reference spectra (1-5%) or the real-time detection limit (0.5-20 ppb), whichever was larger. Typical stack concentrations
ranged from hundreds of ppb to thousands of ppm depending on the analyte (Fig. 1 and Stockwell et al., 2014).

**2.3.2 Photoacoustic extinctiometers (PAX) at 870 and 401 nm**

Particle absorption and scattering coefficients ($B_{abs}$, Mm$^{-1}$, $B_{scat}$, Mm$^{-1}$) were measured directly at 1 s time resolution using two

photoacoustic extinctiometers (PAX, Droplet Measurement Technologies, Inc., Longmont, CO), and single scattering albedo
(SSA) at 401 and 870 nm, and the Ångström absorption exponent (AAE) were derived using those measurements. A 1L min$^{-1}$
aerosol sample flow was drawn through each PAX using a downstream pump and split internally between a nephelometer and
photoacoustic resonator for simultaneous measurement of light scattering and absorption. Scattering of the PAX laser was
measured using a wide-angle reciprocal nephelometer that responds to all particle types regardless of chemical makeup, mixing

state, or morphology. For absorption measurements, the laser beam was directed through the aerosol stream and modulated at a
resonant frequency of the acoustic chamber. Absorbing particles transferred heat to the surrounding air, inducing pressure waves
that were detected via a sensitive microphone. Advantages of the PAX include direct in-situ measurements, a fast response time,
continuous autonomous operation, and eliminating the need for filter collection and the uncertainties that come with filter
artifacts (Subramanian et al., 2007).

We sampled stack burns through ~2 m of 0.638 cm (o.d.) Cu tubing that ran from the stack to a splitter that connected the two
instruments. From the splitter, each separate sample line encountered a scrubber (Purafil-SP Media) to remove absorbing gases
and then a drier (Silica Gel 4-10 mesh) to remove water, with this order ensuring that both instruments were sampling at the
same relative humidity (varying between 13 and 30%). After the drier, each sample line featured a 1.0 μm size-cutoff cyclone
and two acoustic notch filters that reduced noise. Both PAX instruments were calibrated before and after the experiment using

the manufacturer-recommended scattering and absorption calibration procedures utilizing ammonium sulfate particles and a
propane torch to generate pure scattering and strongly absorbing aerosols, respectively. The estimated uncertainty in PAX
absorption and scattering measurements has been estimated as ~4–11% (Nakayama et al., 2015).



### 2.4 Emission ratios (ER) and emission factors (EF) and modified combustion efficiency (MCE)

We convert the time series of mixing ratios for each analyte (Fig. 1) into a form that is broadly useful to others for implementation in local to global chemistry and climate models. For this, we produce emissions ratios (ER) and emission factors (EF). The process starts by calculating excess mixing ratios (denoted $\Delta X$ for each species "X") for all 22 gas-phase species measured using OP-FTIR by subtracting the relatively small average background mixing ratio measured before each fire from all the mixing ratios observed during the burn. The molar emission ratio (ER) for each species X relative to $CO_2$ ($\Delta X / \Delta CO_2$) is the ratio between the sum of the $\Delta X$ over the entire fire relative to the sum of the $\Delta CO_2$ over the entire fire. A comparison of the sums is valid because the large entrainment flow ensures a constant total flow. Molar ER-to-$CO_2$ ratios were calculated for all the species measured using OP-FTIR for all 75 stack burns and the two most important room burns. For the other room burns, OP-FTIR data was generated only for $CO_2$, $CO$, $CH_4$, and $H_2O$ as losses in the room add uncertainty to the mixing ratios for NMOG, $NH_3$, etc. The emission ratios to $CO_2$ were then used to derive EFs calculated by the carbon mass balance method (CMB), which assumes all of the burned carbon is volatilized and that all of the major carbon-containing species have been measured (Ward and Radke, 1993; Yokelson et al., 1996, 1999; Burling et al., 2010, Stockwell et al., 2014):

$$EF(X)\left(g\,kg^{-1}\right) = F_C \times 1000 \times \frac{MM_x}{AM_C} \times \frac{\dfrac{\Delta X}{\Delta CO_2}}{\sum_{j=1}^{n}\left(NC_j \times \dfrac{\Delta C_j}{\Delta CO_2}\right)} \qquad (1)$$

where $F_C$ is the measured carbon mass fraction of the fuel; $MM_x$ is the molar mass of species X; $AM_C$ is the atomic mass of carbon (12 g mol$^{-1}$); $NC_j$ is the number of carbon atoms in each species j; $\Delta C_j$ or $\Delta X$ referenced to $\Delta CO_2$ are the source-average molar emission ratios for the respective species. The denominator of the last term in Eq. (1) estimates total carbon. Based on many BB combustion sources measured in the past, the species $CO_2$, $CO$, and $CH_4$ usually comprise 97-99% of the total carbon emissions (Akagi et al., 2011; Stockwell et al., 2015). Our estimate of total carbon in this paper includes these three species and all the rest of the C-containing emissions measured by the OP-FTIR and the PAXs. Samples of each fuel component were analyzed for moisture content by weighing until dry and for C, H, N, S, and Cl by a commercial (ALS, Tucson) and an academic lab, whose results agreed well with each other on several overlapping fuel samples. The fire-average carbon mass fractions for mixed fuel beds were calculated from the average of the relevant fuel component analyses weighted by dry mass (Tab. S1). The usually small error in the CMB caused by neglecting char formation (Bertschi et al 2003) tends to be canceled by more complete combustion of the higher-C components (Santín et al., 2015) and both these effects are ignored here, but explored in more detail in a companion study (Santín et al., in preparation).

Two major combustion processes are often recognized for open burning of biomass: flaming and smoldering. Combustion efficiency is the fraction of fuel carbon converted to carbon as $CO_2$, which is maximized by flaming combustion, but the modified combustion efficiency (MCE) is also a useful approach for characterizing the relative amount of smoldering and flaming combustion by comparing the fuel carbon converted to $CO_2$ versus $CO_2 + CO$. Although the two processes often occur simultaneously throughout a fire, a high MCE (near 0.99) is an indication of nearly pure flaming, while a lower MCE (~0.8) is an indication of nearly pure smoldering (Akagi et al., 2011):

$$MCE = \frac{\Delta CO_2}{\Delta CO + \Delta CO_2} \qquad (2)$$





In the PAX, the 870 nm laser is absorbed by in-situ black carbon containing particles only without filter or filter-loading effects that can be difficult to correct. We directly measure aerosol absorption ($B_{abs}$, Mm⁻¹... ) — $B_{abs}$, $Mm^{-1}$ — and used the literature-recommended mass absorption coefficient (MAC) ($4.74 \pm 0.63$ $m^2$ $g^{-1}$ at 870 nm) to calculate the BC concentration ($\mu g$ $m^{-3}$) (Bond and Bergstrom, 2006), but the BC mass can be adjusted using different MAC values if supported by future work. Because the PAXs also measured light scattering, scattering and absorption values can be combined to directly calculate the single scattering albedo

(SSA, the ratio of scattering to total extinction). SSA is a useful input for climate models, where an SSA closer to 1 indicates a more "cooling" highly-scattering aerosol:

$$SSA = \frac{Scattering\ (\lambda)}{Scattering\ (\lambda) + Absorption\ (\lambda)} \qquad (3)$$

To a good approximation, $sp^2$-hybridized carbon (including BC) absorbs light proportional to frequency (Bond and Bergstrom, 2006). Thus, the $B_{abs}$ contribution from BC at 401 nm can be calculated from 2.17 times $B_{abs}$ at 870, and any additional $B_{abs}$ at 401 can be assigned to BrC subject to limitations due to "lensing" by coatings discussed elsewhere (Pokhrel et al., 2016; 2017; Lack and Langridge (2013)). Pokhrel et al. (2017) found that coatings typically accounted for much less than 30% of absorption in room burn smoke 1-2 hours old and coatings are likely much less important in 5 s old stack burn smoke (Akagi et al., 2012).

Coating effects are very difficult to deconvolve from BrC effects even with additional instruments that were not available during the stack burns (Pokhrel et al., 2017). This adds some uncertainty to the BrC attribution, but not to the absorption measurements themselves. Absorption by the BrC component of OA means that an approximate mass of OA can be calculated using an OA MAC of $0.98 m^2/g$ (Lack and Langridge, 2013), but the MAC for OA is variable because BrC chemistry and BrC/OA vary and the OA MAC is also highly dependent on the BC/OA ratio as described elsewhere (Saleh et al., 2014). We use the qualitative

OA as a small term in our CMB that helps account for unmeasured C-species, but do not report it in the tables as a quantitative species. Critically though, we do report the OA absorption due to BrC at 401 nm, a poorly characterized term that needs to be improved in climate models to better estimate the radiative forcing of BB aerosol (Feng et al., 2013). The mass ratio of BC to the simultaneous co-located $CO_2$, measured by the FTIR, was multiplied by EF for $CO_2$ to determine mass EFs for BC (g kg⁻¹). The EFs for scattering and absorption at 870 and 401 nm (EF $B_{abs}$, EF $B_{scat}$) were calculated from the measured ratios of $B_{abs}$ and $B_{scat}$

to $CO_2$ and reported in units of $m^2$ per kg of dry fuel burned. We also report the portion of $B_{abs}$ at 401 nm due to BrC. Finally, the Ångström absorption exponent (AAE) (401/870) can be calculated from the 401 and 870 data, where the AAE of pure BC is close to one and larger values are indicative of smoke absorption more dominated by BrC emissions:

$$AAE = -\frac{\log\left(\frac{B_{abs,1}}{B_{abs,2}}\right)}{\log\left(\frac{\lambda_1}{\lambda_2}\right)} \qquad (4)$$

The AAE is useful as an indicator of BrC/BC, but in addition, the full aerosol absorption spectrum is often approximated with a power law function (absorption = C × $\lambda^{-AAE}$) and thus the AAE determined with any wavelength pair can be used to approximately calculate the shape of absorption across the UV-VIS range (Reid et al., 2005).



## 3 Results and Discussion

### 3.1 Overview of wildfire trace gas emissions

We sampled a total of 75 stack burns and 32 room burns at the FSL combustion facility during October and November 2016. Figure 1 shows temporal profiles for the excess mixing ratios of 21 gas-phase compounds (not including water) measured by OP-FTIR for a complete Juniper canopy fire (fire 75). Immediately after ignition, the fire is characterized by a large increase in $CO_2$, corresponding to flaming, followed by a slower increase in CO from smoldering combustion. As is common for most fires, there is no clear distinction between flaming and smoldering but rather an evolving mix of the two processes. Fire-integrated ERs to $CO_2$ and EFs were determined for all 75 stack fires based on the whole fire. For room burns, we calculated EF based on integrating the $\Delta X$ only up to the point where emissions were well mixed to capture the whole fire, but also minimize the effect of wall losses and infiltration (see Fig. 3 in Stockwell et al., 2014). The fire-integrated EFs for some of the most common Western U.S ecosystem fuel complexes sampled in this study are summarized in Table 1. These are averages of the replicate fires (three to four replicate measurements for each fuel type). Table 1 does not reveal a strong ecosystem dependence across the coniferous ecosystems, but does indicate lower EFs for many pollutants emitted by the chaparral fires. However, large wildfires often burn in multiple fuel types simultaneously. For instance, the Rim Fire burned in pine, pine-oak, and chaparral fuels simultaneously (Liu et al., 2017). These factors justify using a single set of EFs for all wildfires, unless detailed fuels data is available that warrants more precise EF estimates. The derivation of the best wildfire EFs is explored in more detail in the next section. A summary of all the EFs we measured by OP-FTIR and PAX can be found in Table S2, with the averages of specific fuel components and complexes found in Table S3. Numerous additional NMOG that were measured by other instruments (e.g. $H_3O^+$ CIMS and $I^-$ CIMS) will be presented elsewhere (Koss et al., in preparation). These additional species are often reactive and very important in plume chemistry even though they have only a small effect on the carbon mass balance. A few species were measured by both OP-FTIR and MS and the preferred values depend on several issues such as S:N (often better on MS), interference (often worse on MS), "stickiness," fragmentation, and proton affinity that are discussed in more detail elsewhere (Koss et al., in preparation).

### 3.2 Comparison of lab EF to wildfire EF

It is important to compare our FIREX lab fire emissions data to field measurements of real wildfires to assess how representative and useful the lab-based data are, especially for the many species measured in the lab, but not the field. EF measurements on real wildfires are rare, but Liu et al. (2017) report recent EFs for three wildfires sampled during the 2013 Studies of Emissions and Atmospheric Compositions, Clouds, and Climate Coupling by Regional Surveys (SEAC⁴RS, https://espo.nasa.gov/missions/seac4rs) (Toon et al., 2016) campaign, and the Biomass Burning Observation Project (BBOP, https://www.arm.gov/research/campaigns/aaf2013bbop) campaign.

We compare our results from the FSL combustion studies to those reported by Liu et al. in two ways. In method one, we plot the lab-measured EFs against their corresponding MCEs for all the fires and we fit the data with a linear regression relationship for each compound. Using the slope and y-intercept of the linear regression, and the field average MCE from Liu et al. of 0.912, we calculate a lab-based prediction of EF at the field-average MCE for each compound measured by OP-FTIR. Fig. 2 shows an example of the procedure for $CH_4$, comparing the lab-predicted EF at the field-average MCE to the average field-measured wildfire EF. In method 2, we compared the simple lab-average EF to the average field–measured wildfire EF. The results of these two methods are shown in Tab. 2 and Fig. 2. Method one is generally preferred because the lab fires had higher average MCE (i.e. a higher fire-integrated flaming/smoldering ratio) than the real wildfires sampled to date, most likely due to some unavoidable drying of the fuels during storage. The differences between the lab prediction at the field average MCE and the field





average emissions are probably mostly due to the relative age of the smoke and the reactivity of compounds. The field study
       included smoke samples up to two hours old and elevated OH, $HO_2$, $H_2O_2$, $O_3$, etc. have been observed in fresh smoke plumes
       (Hobbs et al., 2003; Yokelson et al., 2009) so the more reactive species (e.g. $SO_2$, HCl, $NO_x$, and some NMOG) have lower EF
       in the field data. For example, the lab/field ratio increases going from ethylene to propene, to 1,3-butadiene in accordance with,
       though not directly proportional to, their increasing OH rate constants; and other chemistry, instrumental, and sampling
challenges are relevant for some species (e.g. Finlayson-Pitts and Pitts, 2000; Apel et al., 2003; Fig. 7 in Hornbrook et al., 2011;
       Burkholder et al., 2015). A few reactive species were measured in two older airborne studies of fresh western US wildfire smoke
       and they agree significantly better with our lab-based predictions (Radke et al., 1991; Hobbs et al., 1996). For instance, Radke et
       al. (1991) report EFs for $NO_x$ as NO (2.0 g/kg), $NH_3$ (2.0 g/kg), and $C_3H_6$ (0.70 g/kg) for the Myrtle Fall Creek wildfire that are
       all within 20% of our lab-predicted EFs. Hobbs et al. (1996) report an EF for $SO_2$ (0.79 g/kg) that is closer to our value than the
Liu et al. value is despite the much lower MCE (0.81).
       Fig. 3 shows the comparison for method one from Tab. 2 graphically. From Fig. 3 it is clear that for the main relatively stable
       compounds, including formaldehyde, methanol, and hydroxyacetone; the lab-predicted EF falls within 20% of the measured
       wildfire EF and all the emissions except $NO_x$ and $SO_2$ overlap within the observed variability. Also highlighted in Figure 3;
       many compounds such as HONO, acetic acid, ammonia, phenol, glycolaldehyde, formic acid, etc. were measured only for our
lab fires. The lab-measured EFs for these OP-FTIR species and the data for many NMOG species measured by MS and FIREX
       data in general can thus be used to generate representative EFs or other data for real wildfires. Many of these EFs are critically
       important to represent wildfire emissions well: e.g. $NH_3$ (Benedict et al., 2017) and SOA or PAN precursors (Alvarado et al.,
       2015; Müller et al., 2016). Other approaches to generate representative data that are not explored in detail here, but should work
       well include reporting the average for the lab fires clustered around the field average MCE (Fires 8, 39, 45, 51, 59, and 66) or
reporting ER to CO (e.g. Koss et al., in preparation), where the latter can also be converted to EF by coupling with the field
       average EFCO. For example, if we take the average of six fires clustered around the field average MCE in the $CH_4$ plot shown in
       Figure 2, we get an average EF for $CH_4$ of 4.69, which is close to the Liu et al., reported value of 4.90. Alternatively, we can
       calculate a molar ER for $CH_4$ to CO for all the lab fires (0.071), then utilize the wildfire average EF CO reported by Liu et al
       (89.3 g $kg^{-1}$) to calculate a new EF. Using this method, we get an EF for $CH_4$ of 3.78, which is within 20% of the field average
value. Either of these methods should help reflect the field average flaming/smoldering ratio. In addition, positive matrix
       factorization was found to be a very useful method to predict field EF from the lab data for NMOG as discussed elsewhere
       (Sekimoto et al., in preparation). Finally, given the small amount of field sampling, more field work is clearly needed.

### 3.3 EF dependence on fuel

We burned individual fuel components (duff, litter, canopy, etc.) in addition to mixtures of all the major components found in
       widespread Western US coniferous ecosystems for insights into fuel effects on emissions and to what degree specific emissions
       were enhanced by a certain component. For example, Figure 4 shows the EFs of 21 trace gases from the Douglas fir ecosystem
       fuel mixture burns side by side with the EFs from burning pure Douglas fir components in separate fires. Emissions of
       furaldehyde, formaldehyde and methanol were enhanced when burning a pure rotten log component; while acetylene, ethylene,
and propene, as well as other non-methane hydrocarbons (NMHC), were more prevalent in emissions from Douglas fir canopy.
       We did the same analysis for a Ponderosa pine ecosystem (Figure 5). While the canopy component in Ponderosa pine produced
       enhanced emissions of NMHC, the rotten log did not contribute to the same level of enhancement in furaldehyde, formaldehyde
       and methanol. However, for both vegetation types we observed an enhancement in $NO_x$ emissions from the litter and canopy
       components, which is likely due to high nitrogen content (0.69% N for litter, 0.81% N for canopy) and the tendency to burn by





flaming for these components. Mixed coniferous ecosystem values are fairly similar for both fuels and agree within 30% for the majority of compounds, excluding methanol, furan, and $NO_x$. We also note that while the mixed Douglas fir and Ponderosa pine ecosystems that we burned contained canopy, litter, and woody components in varying diameter classes, they did not contain a rotten log since the latter component is not included in FOFEM. We further investigate fuel variability by taking pure components from several ecosystems and comparing them to one another. Figure 6 shows species emitted by duff from three

different coniferous ecosystems. Acetic acid and methanol are strongly emitted by all three duff fuels, but ammonia enhancement occurs in only Engelmann spruce and Subalpine fir fuels. Jeffrey pine duff had a lower EF for $NH_3$ despite similar fuel N. This could possibly be due to the age of the fuel as it was contained in storage longer than other fuels and not fresh. Additional results for other fuel components (rotten log, canopy, litter) are in Figures S1, S2, and S3 respectively.

**3.4 Overview of optical properties**

As mentioned previously, we measured absorption and scattering coefficients directly at 401 and 870 nm. For the first 31 stack fires, which includes most of the studied fuel types, we have both 401 and 870 nm data. For the remaining 44 stack fires, we only report data at 870 nm as we used our 401 nm PAX for intercomparison studies that will be reported elsewhere. Figure 7 plots the AAE and SSA at both wavelengths of 31 stack fires as a function of MCE. High AAE is an indicator of BrC and relates to

smoldering, which is denoted by low MCE and high SSA values. Low AAE, along with low SSA and high MCE values, indicates more flaming combustion. The lab-based average fire-integrated optical properties for some of the most common Western U.S ecosystems are listed in Table 3. Table 3 does not reveal a strong ecosystem dependence among coniferous ecosystems tested for optical properties, but does indicate that chaparral fire aerosol is relatively more absorbing and that there are significant contributions of absorption by BrC at 401 nm among all ecosystems. The absorption by BrC is responsible for at

least half and up to two thirds of the absorption at 401 nm even at higher MCE. The lab average AAE of $2.19 \pm 0.24$ across all 31 fires confirms a role for BrC, while the lab-average SSA at both wavelengths indicates the fresh aerosol has a net warming influence in the atmosphere (SSA < 0.9, Praveen et al., 2012); although SSA can increase with smoke age (Yokelson et al., 2009). The absorption of BrC at 401 nm has several implications in atmospheric chemistry, including impacts on UV-driven photochemical reactions producing ozone, and the lifetime of $NO_x$ and HONO. Furthermore, because of its absorbing nature,

factoring in the BrC could mean the net radiative forcing of biomass burning is not cooling or neutral as often assumed, but warming if the BrC is sufficiently long-lived as probed in other FIREX studies and previous papers (e.g. Feng et al., 2013; Forrister et al., 2015).

**3.5 Comparison of lab optical properties to field optical properties**

There are very few field measurements of the optical properties of smoke from US wildfires, but we can compare our results from the lab studies to the initial aerosol optical properties for one wildfire (the Rim Fire) reported by Liu et al. and Forrister et al. (2015). An AAE of 3.75 at an MCE of 0.923 for the Rim Fire is reported between these two studies. With the linear regression of the lab data shown in Figure 7, we can predict an AAE of 3.31 at the wildfire field average MCE (0.912) and an AAE of 2.91 at the Rim Fire MCE (0.923) using prediction method one described in Section 3.2. At the wildfire field average

MCE, our calculated AAE represents 88% of the reported Rim Fire AAE, while at the Rim Fire MCE, our calculated AAE represents 78% of the reported Rim Fire AAE. Although our calculated values are relatively close to the reported value, a small change in AAE implies a big change in the BrC/BC absorption ratio, but only a small change in the % absorption by BrC. Our AAE values imply that BrC accounts for 77 to 82% of the absorption at 401. The average of the AAE from the single Rim Fire





measurement (3.75) and the AAE predicted from the more extensive lab fires (3.31) is ~3.5, which may be a reasonable best
    guess at the AAE of fresh US wildfire smoke and implies that ~86% of absorption at 401 nm is due to BrC.

    In Figure 8, we plot the initial % absorption by BrC at 401 nm for the Rim Fire measured AAE and for our lab-estimated AAE at
    the field average MCE. Figure 8 also shows the lab-measured total $EF_{abs}401$ and the BrC contribution to $EF_{abs}401$ for 31 lab
    fires. BrC dominates absorption at 401 nm at low MCE values and as MCE increases, BrC absorption remains a significant but
variable component of overall absorption. The variability is likely due to realistic "natural" fire-to-fire variability in fuels,
    moisture content, etc.

    In Table 4 we report the study-averages for BC mass EF, absorption and scattering EFs, SSA, and AAE. The quantities that
    require 401 nm data are averages for the 31 stack fires where 401 and 870 nm data were obtained, while the quantities that need
    just 870 nm data are averages for all 75 stack fires. Thus, a few of the values may differ from Table 2 and 3, which are based
only on the first 31 fires. We also show the comparison of our lab-average and lab-predicted AAEs to the AAE in Liu et al.,
    (2017) and our lab-average and lab-predicted BC EF to the unpublished BC EF calculated as part of Liu et al., (2017). Table 4
    also presents a set of equations that can be used to fit lab-measured optical properties and make predictions at any MCE.
    However, more measurements of wildfires in the field and the lab (including aging) are needed to asses wildfire aerosol optical
    properties.

### 3.6 Fuel dependence of aerosol optical properties

    Burning individual fuel components in addition to mixtures found in typical, widespread western U.S ecosystems allows us to
    investigate the extent to which optical properties are either enhanced or diminished by certain components. Table 5 lists the
    study-average BC EF and optical properties for all the coniferous ecosystems shown in Table 3 and the study-average BC EF and
optical properties for the individual fuel components averaged across all the coniferous ecosystems. The averages and standard
    deviations for each reported quantity indicate that there is large variation among specific components and a large coefficient of
    variation for the coniferous ecosystem average. The variability could potentially depend on ecosystem type, fuel components,
    fuel moisture, or other things as discussed for trace gases in section 3.3. While there is considerable variation within each
    ecosystem type, the individual ecosystem averages in Table 3 all agree within 38% of the study-average for all the coniferous
ecosystems shown in Table 5 and the AAEs are all within 20%. However, Table 5 also shows that the average AAE for some
    ecosystem components is very different from the average AAE for all the coniferous ecosystems (2.26). For instance, the largest
    contribution to a high AAE per fuel component consumed comes from duff, where BrC accounts for almost all of the absorption
    at 401 nm (AAE 7.13). The rotten log component also contributes an anomalously high average AAE of 4.60.   Thus, these
    components contribute more BrC relative to BC in proportion to their fuel consumption to the mixed ecosystem results where
AAE is 2.26 and BrC accounts for just over half of the absorption at 401 nm. Conversely, litter consumption would tend to lower
    a fuel mixture's AAE. However, AAE is a measure of the shape of the aerosol absorption cross-section and the absorption EFs
    are a measure of total emissions of absorbing material. In this respect, litter produces more BC absorption and more BrC
    absorption per unit mass than duff though at a lower BrC/BC ratio than duff. This is consistent with the lower SSA for litter. We
    conclude that the variability in mixed ecosystem optical properties was likely due to variable consumption of pure components,
with a weaker dependence on the dominant tree species. For example, much of the variability in ecosystem average AAEs and
    the study average AAE is linked to the varying amount of duff consumed in the mixed fuel beds (Table S1). (The variability in
    actual duff consumption is likely larger than the variability in duff loading shown as the amount of residual material also varied.)
    Duff consumption in the field is increased by drought conditions, which would contribute variability on real fires (Davies et al.,
    2013). In summary, the results presented indicate that, in all cases, the overall ecosystem and mixture of components produces a




significant amount of BrC. As mentioned previously, this has several implications in regional atmospheric chemistry and radiative forcing. Additional instruments were deployed on room burn experiments, where the fuels were also purposely changed to investigate the effect on optical properties and will be reported elsewhere.

**3.7 Trace gases and BC emissions of peat, dung, and rice straw combustion**

We also measured emissions from several fires of peat, rice straw, and dung, due to their widespread burning in Asia and their value as extreme examples of fuel impacts (e.g. high smoldering/flaming or high N or Cl content). Peat, which is especially important in Southeast Asia (Stockwell et al., 2016a) is similar to duff found in the western U.S in that it is often consumed by pure smoldering combustion and has a high AAE, high HCN emissions, and low BC. Although we did not measure AAE for peat, we do report an MCE of 0.83, where a low MCE likely indicates a high AAE. We also report EF's for $CH_4$ (10.83 g $kg^{-1}$),

HCN (3.97 g $kg^{-1}$), acetic acid (4.44 g $kg^{-1}$) and BC (0.003 g $kg^{-1}$). We compare these values to the field measurements reported in Stockwell et al. (2016a): $CH_4$ (9.51 ± 4.74 g $kg^{-1}$), HCN (5.75 ± 1.60 g $kg^{-1}$), acetic acid (3.89 ± 1.65 g $kg^{-1}$) and BC (0.006 ± 0.002 g $kg^{-1}$) and find that our values agree well (BC extremely small and gases within 31%) between peat measured in the lab and peat measured in the field. (A more detailed comparison will follow planned field measurements.)

Additionally, we compare our dung MCE value (0.90), $CH_4$ (6.63 g $kg^{-1}$), HCN (1.96 g $kg^{-1}$), acetic acid (6.36 g $kg^{-1}$), and BC

(0.01 g $kg^{-1}$) values to those based on field work in Nepal reported in Stockwell et al. (2016b): MCE (0.90), $CH_4$ (6.65 ± 0.46 g $kg^{-1}$), HCN (2.01 ± 1.25 g $kg^{-1}$), acetic acid (7.32 ± 6.59 g $kg^{-1}$), and BC (0.004 ± 0.003 g $kg^{-1}$). We find excellent agreement between our values (15% for trace gases and BC very small) and those reported from field measurements in Nepal.

Rice straw was burned because of its global importance in agricultural waste burning and to probe the extremes of fuel chemistry (Akagi et al., 2011). Grasses are usually very high in chlorine content (0.61%, Table S1; Lobert et al., 1996) and our EF for HCl

of 0.65 g $kg^{-1}$ for rice straw was the highest of any fuel measured during the FIREX campaign. Furthermore, our rice straw EF for HCl is comparable to Stockwell et al. 2015 (0.43 ± 0.29). The findings summarized in this section further suggest and reinforce the idea that simulated lab fires can probe fuel effects and provide an accurate representation of measurements in the field, even outside the scope of Western U.S. wildfires.

**4. Conclusions**

We measured trace gas and aerosol emissions from 107 simulated western wildfires during the FIREX campaign in the fall of 2016 using OP-FTIR and PAX. For 31 stack fires, we report aerosol measurements based on both 401 and 870 nm, and for the remaining 44 stack fires we report aerosol characteristics based on only 870 nm data. We provide the MCE and the mass EF (g $kg^{-1}$) for 23 different trace gases (not including water) and BC. We also provide the scattering and absorption EF ($m^2$ $kg^{-1}$) at 870

and 401 nm along with the $EF_{abs}$401 due to brown carbon only, SSA, and AAE. We burned canopy, litter, duff, dead wood, and other fuels in combinations using FOFEM to represent relevant ecosystems and as pure components to investigate the effects of individual fuels. Full trace gas data are reported for all 75 stack burns and two room burns, and $CO_2$, CO, $CH_4$ and MCE were archived for the remaining room burns. We found little variability in average trace gas EFs across coniferous ecosystems, but the average EFs for two chaparral species were similar to each other and lower than in coniferous ecosystems for most pollutants,

including $CH_4$ (1.20 ± 0.09 g $kg^{-1}$), formaldehyde (0.50 ± 0.06 g $kg^{-1}$), glycolaldehyde (0.15 g $kg^{-1}$) and HCN (0.09 g $kg^{-1}$) to name a few. Additionally, there was considerable variability in the average trace gas EF for certain fuel components. For instance, emissions of NMHC were enhanced from rotten logs and emissions of $NO_x$ were enhanced from litter and canopy components.



In similar fashion, there was little variation in the average optical properties for the different mixed coniferous ecosystems, but individual fuel components like duff and rotten logs contributed significantly on a per mass basis to the relative importance of BrC and BC, with BrC accounting for nearly 100% and 78% of the absorption at 401, respectively, for these fuel components. The lab-average AAE for all 31 fires, including those burning components like chaparral and coniferous canopy, which tend to burn more by flaming, was 2.8 (Tab. 4) indicating that BrC absorption contributed to over half (64%) of the absorption at 401

nm for the lab fires on average.

We compared the trace gas and aerosol emissions from the fires in our laboratory-simulated Western U.S ecosystems to those from real Western U.S wildfires measured in slightly-aged smoke in the field as reported by Liu et al. (2017) and Forrister et al. (2015). Using a simple procedure to account for the flaming to smoldering ratio, we generated EF values from the lab data that were in agreement with the field data for most "stable" trace gases, including $CH_4$ (within 3%), formaldehyde (within 4%),

methanol (within 20%), and hydroxyacetone (within 1% agreement). Most of the EF discrepancies were due to the field smoke being more aged. The excellent agreement suggests that FIREX data can be confidently used in general to represent real fires; especially for species not measured yet in the field. For instance, important compounds not previously measured in the field for western wildfires, but measured in this study include ammonia (1.65 g kg$^{-1}$), acetic acid (2.44 g kg$^{-1}$), HONO, and others (Fig. 3). Optical properties were not compared as extensively because limited field data are available, which highlights the need for more

field measurements on true wildfires. However, a preliminary best guess for a fresh wildfire smoke AAE of ~3.5 is supported by averaging the lab-based predictions and more limited field data. Impacts on photochemical reactions producing ozone, and the lifetime of $NO_x$ and HONO are likely as a result of the strong abundance of BrC. In addition, recognizing the presence of absorbing BrC in biomass burning plumes could alter the modeled contribution of biomass burning to net radiative forcing in a more positive direction. Finally, to investigate fuel chemistry impacts and due to their widespread global importance, we also

measured EFs for fires in peat, dung, and rice straw and compared to field values reported by Stockwell et al. (2015, 2016a, 2016b). Our lab-based EFs for all three of these fuels were in good agreement with the field studies. Overall, our lab-simulated fires can provide important emissions data that is fairly representative of real fires and used to accurately assess BB impacts.

Data Availability:

Raw data used to derive the EFs and other quantities reported that are not included in the supplemental information or the NOAA archive can be obtained by contacting the corresponding author.

Author Contributions

VS, RY, JMR, CW, JdG, and JR designed the research. VS, RY, and DG performed the measurements and/or contributed to the

data analysis. All authors contributed to the discussion and interpretation of the results and writing the manuscript.

Competing Interests

The authors declare that they have no competing interests.

*Acknowledgements*:

Vanessa Selimovic and Robert Yokelson were supported by NOAA-CPO grant NA16OAR4310100. Indonesian peat was provided through NASA grant NNX13AP46G to UM. Purchase and preparation of the PAXs was supported by NSF grant AGS-1349976 to R. Y. Parts of this work were supported by NOAA's Climate and Health of the Atmosphere initiatives. We would also like to extend our thanks to Ted Christian, Edward O'Donnell, Maegan Dills, Roger Ottmar, David Weise, Mark Cochrane,





Kevin Ryan, and Robert Keane for help with fuels and related assistance, and Shawn Urbanski, and Thomas Dzomba for logistics assistance. Joost de Gouw worked as a consultant for Aerodyne Research Inc. during part of the preparation phase of this manuscript. We thank the NOAA BC group for the Rim Fire BC data and Xiaoxi Liu for sharing her calculation of the Rim Fire EFBC.

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

35

40





5    Table 1. Average emission factors (g kg$^{-1}$) of common Western U.S ecosystems measured in the lab.

| Compound | Douglas Fir | Engelmann Spruce | Lodgepole Pine | Ponderosa Pine | Subalpine Fir | Chaparral - Chamise | Chaparral - Manzanita |
|---|---|---|---|---|---|---|---|
| $CO_2$ | 1682.95 (25.03) | 1642.88 (57.76) | 1686.22 (23.02) | 1696.45 (23.00) | 1659.79 (10.91) | 1713.19 (17.40) | 1696.17 (19.02) |
| CO | 65.74 (12.59) | 69.33 (18.37) | 70.42 (9.66) | 78.41 (10.91) | 72.80 (5.07) | 55.88 (4.96) | 40.56 (0.64) |
| $CH_4$ | 2.31 (0.39) | 3.02 (1.38) | 2.61 (0.32) | 2.76 (0.85) | 3.86 (1.34) | 1.26 (0.10) | 1.14 (0.07) |
| Methanol ($CH_3OH$) | 0.73 (0.14) | 1.34 (0.70) | 0.86 (0.20) | 1.31 (0.59) | 1.28 (0.55) | 0.40 (0.04) | 0.53 (0.07) |
| Formaldehyde (HCHO) | 1.52 (0.40) | 1.56 (0.26) | 1.67 (0.50) | 1.79 (0.46) | 1.92 (0.32) | 0.54 (0.003) | 0.46 (0.14) |
| Hydrochloric Acid (HCl) | -- | 0.05 | -- | -- | -- | -- | -- |
| Acetylene ($C_2H_2$) | 0.40 (0.11) | 0.32 (0.07) | 0.55 (0.11) | 0.47 (0.15) | 0.50 (0.05) | 0.31 (0.08) | 0.21 (0.09) |
| Ethylene ($C_2H_4$) | 1.33 (0.31) | 1.18 (0.21) | 1.85 (0.35) | 1.61 (0.47) | 1.86 (0.53) | 0.48 (0.05) | 0.56 (0.18) |
| Propene ($C_3H_6$) | 0.35 (0.05) | 0.45 (0.20) | 0.70 (0.42) | 0.51 (0.14) | 0.68 (0.36) | 0.11 (0.01) | 0.17 (0.05) |
| Ammonia ($NH_3$) | 0.47 (0.07) | 1.13 (0.70) | 0.62 (0.13) | 0.69 (0.22) | 0.85 (0.57) | 0.56 (0.02) | 0.52 (0.03) |
| 1,3-Butadiene | 0.01 | 0.02 | 0.07 (0.02) | 0.04 (0.02) | 0.09 (0.03) | -- | -- |
| Acetic Acid ($CH_3COOH$) | 1.14 (0.20) | 1.71 (0.46) | 1.12 (0.46) | 1.63 (1.03) | 1.99 (1.36) | 0.74 (0.05) | 1.74 (1.39) |
| Formic Acid ($CH_2O_2$) | 0.25 (0.06) | 0.23 (0.02) | 0.21 (0.05) | 0.28 (0.09) | 0.26 (0.06) | 0.05 (0.002) | 0.18 (0.16) |
| Furan ($C_4H_4O$) | 0.14 (0.05) | 0.15 (0.11) | 0.18 (0.04) | 0.30 (0.10) | 0.16 (0.03) | 0.06 (0.03) | 0.46 (0.59) |
| Hydroxyacetone | 0.58 (0.06) | 0.75 (0.16) | 0.52 (0.29) | 0.97 (0.29) | 0.72 (0.09) | 0.36 (0.07) | 0.31 (0.08) |
| Phenol | 0.46 (0.41) | 0.62 (0.09) | 0.42 (0.18) | 0.89 (0.20) | 0.61 (0.27) | 0.49 (0.07) | 0.31 (0.09) |
| Furaldehyde | 0.68 | 0.72 (0.17) | 0.73 (0.06) | 0.95 (0.26) | 0.58 (0.37) | 0.53 (0.25) | 0.72 (0.11) |
| NO | 1.83 (0.24) | 1.71 (0.11) | 1.84 (0.14) | 1.25 (0.40) | 1.85 (0.12) | 2.39 (0.05) | 1.89 (0.01) |
| $NO_2$ | 1.57 (0.32) | 2.03 (0.44) | 1.13 (0.32) | 1.53 (0.70) | 1.37 (0.19) | 0.49 (0.11) | 0.81 (0.10) |
| HONO | 0.65 | 0.42 (0.16) | 0.68 (0.05) | 0.6 (0.19) | 0.71 | 0.48 (0.11) | 0.44 (0.01) |





| | | | | | | | |
|---|---|---|---|---|---|---|---|
| | (0.18) | | | | (0.05) | | |
| Glycolaldehyde | 0.53 (0.06) | 0.63 (0.06) | 0.63 (0.10) | 0.69 (0.17) | 0.76 (0.14) | 0.12 | 0.18 |
| HCN | 0.20 (0.02) | 0.27 (0.05) | 0.24 (0.05) | 0.29 (0.08) | 0.25 (0.05) | 0.10 (0.03) | 0.07 |
| SO2 | 1.18 (0.06) | 1.32 (0.19) | 1.31 (0.15) | 1.49 (0.50) | 1.67 (0.48) | 0.82 (0.05) | 0.89 |
| MCE | 0.94 (0.01) | 0.94 (0.02) | 0.94 (0.01) | 0.93 (0.01) | 0.94 (0.01) | 0.95 (0.01) | 0.96 (0.001) |

5    [a]Values in brackets are (1σ) standard deviation.



5    Table 2. Summary of the comparison of emission factors (g kg$^{-1}$) measured in the lab and field.

| Compound | Lab avg EF | Lab eqn slope | Lab eqn intercept | Lab-based Prediction | *Liu et al., 2017* EF | Predicted/ Field | Lab avg/ Field avg |
|---|---|---|---|---|---|---|---|
| $CO_2$ | 1644.70 | 2834.5 | -990.68 | 1594.38 | 1454.00 | 1.10 | 1.13 |
| CO | 78.03 | -1044.9 | 1049.5 | 96.55 | 89.30 | 1.08 | 0.87 |
| $CH_4$ | 3.30 | -81.255 | 78.85 | 4.75 | 4.90 | 0.97 | 0.67 |
| NOx as NO | 2.98 | 22.67 | -18.226 | 2.45 | 0.49 | 5.00 | 6.08 |
| Acetic Acid | 1.88 | -32.221 | 31.825 | 2.44 | -- | -- | -- |
| NO | 1.80 | 12.64 | -10.009 | 1.52 | 0.11 | 13.81 | 16.40 |
| Formaldehyde | 1.66 | -30.44 | 29.959 | 2.20 | 2.29 | 0.96 | 0.72 |
| Ethylene | 1.62 | -16.566 | 17.025 | 1.92 | 0.91 | 2.11 | 1.78 |
| $SO_2$ | 1.37 | -7.8704 | 8.6892 | 1.51 | 0.32 | 4.72 | 4.29 |
| Methanol | 1.31 | -36.258 | 35.02 | 1.95 | 2.45 | 0.80 | 0.54 |
| $NO_2$ | 1.20 | -4.8756 | 5.7599 | 1.31 | 0.58 | 2.26 | 2.07 |
| Ammonia | 1.10 | -31.315 | 30.21 | 1.65 | -- | -- | -- |
| Furaldehyde | 0.82 | -13.849 | 13.703 | 1.07 | -- | -- | -- |
| Hydroxyacetone | 0.80 | -15.888 | 15.617 | 1.13 | 1.13 | 1.00 | 0.71 |
| Glycolaldehyde | 0.72 | -11.346 | 11.259 | 0.91 | -- | -- | -- |
| Phenol | 0.70 | -14.595 | 14.692 | 1.38 | -- | -- | -- |
| Propene | 0.61 | -10.042 | 9.9405 | 0.78 | 0.35 | 2.23 | 1.75 |
| HONO | 0.56 | -2.4264 | 2.8239 | 0.61 | -- | -- | -- |
| Acetylene | 0.45 | -2.4538 | 2.738 | 0.50 | 0.24 | 2.08 | 1.89 |
| HCN | 0.36 | -7.3735 | 7.2029 | 0.48 | 0.34 | 1.41 | 1.05 |
| Formic Acid | 0.27 | -5.3448 | 5.2388 | 0.36 | -- | -- | -- |
| Furan | 0.23 | -5.3508 | 5.2065 | 0.33 | 0.51 | 0.64 | 0.45 |
| 1,3-Butadiene | 0.18 | -9.6611 | 9.1572 | 0.35 | 0.06 | 6.30 | 3.28 |
| HCl | 0.11 | -2.5089 | 2.4628 | 0.17 | 0.004 | 42.5 | 27.5 |
| Average Ratio Smoldering Compounds[a] | | | | | | 0.96 | 0.75 |
| StDev Ratio | | | | | | 0.29 | 0.23 |
| Fractional Uncertainty | | | | | | 0.30 | 0.31 |

[a]Average of less reactive/moderately reactive species: includes formaldehyde, methanol, hydroxyacetone and HCN. Reactive smoldering compounds were left out.



Table 3. Lab-average emission factors ($m^2$ $kg^{-1}$) and fire-integrated optical properties for common Western U.S ecosystems.

| Species | Douglas Fir | Engelmann Spruce | Lodgepole Pine | Ponderosa Pine | Chaparral - Chamise | Chaparral - Manzanita |
|---|---|---|---|---|---|---|
| Black Carbon (g $kg^{-1}$) | 0.23 (0.06) | 0.12 (0.07) | 0.33 (0.14) | 0.48 (0.25) | 0.42 (0.14) | 0.28 (0.03) |
| EF Babs 870 | 1.08 (0.29) | 0.58 (0.32) | 1.59 (0.67) | 2.28 (1.20) | 2.00 (0.68) | 1.32 (0.15) |
| EF Babs 401 | 7.61 (1.10) | 6.21 (0.19) | 10.17 (1.11) | 12.04 (1.07) | 10.37 | 8.63 |
| EF Babs 401 (BrC) | 5.04 (0.70) | 4.40 (0.27) | 5.77 (0.76) | 5.55 (0.76) | 5.56 | 5.54 |
| EF Bscat 870 | 3.01 (1.34) | 3.36 (2.66) | 2.79 (1.40) | 4.55 (1.50) | 0.52 (0.16) | 0.89 (0.51) |
| EF Bscat 401 | 48.31 (7.24) | 62.44 (7.38) | 44.11 (6.99) | 50.19 (9.96) | 11.99 | 23.69 |
| SSA 401 | 0.86 (0.01) | 0.91 (0.01) | 0.82 (0.02) | 0.80 (0.04) | 0.54 | 0.73 |
| SSA 870 | 0.72 (0.08) | 0.82 (0.09) | 0.64 (0.07) | 0.67 (0.11) | 0.21 | 0.39 |
| AAE | 2.43 (0.09) | 2.65 (0.30) | 2.12 (0.19) | 1.84 (0.18) | 2.02 | 2.36 |
| MCE | 0.94 (0.01) | 0.94 (0.02) | 0.94 (0.01) | 0.93 (0.01) | 0.95 (0.01) | 0.96 (0.001) |

[a]Values in brackets are (1σ) standard deviation.





5      Table 4. Summary of the comparison of optical properties and emission factors ($m^2\ kg^{-1}$) measured in lab to the Rim Fire.

| Species | Lab avg EF | Lab eqn | Lab-based prediction using field average MCE | Rim Fire | Predicted/Field | Lab avg/Rim Fire |
|---|---|---|---|---|---|---|
| Black Carbon[b] (g $kg^{-1}$) | 0.53 (0.58) | $y = 1.236x^{26.884}$ | 0.104 | 0.187[e] | 0.56 | 2.83 |
| EF Babs 870[b] | 3.20 (5.16) | $y = 5.86x^{26.884}$ | 0.49 | -- | -- | -- |
| EF Babs 401[c] | 11.11 (5.93) | $y = 11.414x^{1.8236}$ | 9.65 | -- | -- | -- |
| EF Babs 401 (BrC)[c] | 7.11 (5.14) | $y = -31.88x + 36.64$ | 7.57 | -- | -- | -- |
| EF Bscat 870[b] | 10.13 (22.63) | $y = -86.82x + 84.63$ | 5.45 | -- | -- | -- |
| EF Bscat 401[c] | 69.84 (80.20) | $y = -1322.80x + 1295$ | 88.61 | -- | -- | -- |
| SSA (401)[c] | 0.79 (0.13) | -- | 0.90[d] | -- | -- | -- |
| SSA (870)[b] | 0.64 (0.26) | -- | 0.92[d] | -- | -- | -- |
| AAE[c] | 2.80 (1.57) | $y = -35.45x + 35.64$ | 3.31 | 3.75[f] | 0.78 | 0.75 |

[a]Values in brackets are (1σ) standard deviation.
[b]Average for all 75 stack fires where  870 nm data is available.
[c]Average for 31 fires where both 401 and 870 nm is available.
[d]SSA values calculated from $B_{abs}$ and $B_{scat}$ EF

10     [e]Value not published (X. Liu private communication, https://www.nasa.gov/mission_pages/seac4rs/index.html)
[f]From Forrister et al.





5    Table 5. Optical properties and emission factors (m$^2$ kg$^{-1}$) for mixed coniferous ecosystems and ecosystem components.

| Species | Mixed Coniferous Ecosystem[a] | Canopy[b] | Litter[c] | Duff[d] | Rotten Log[e] |
|---|---|---|---|---|---|
| Black Carbon (g kg$^{-1}$) | 0.43 (0.33) | 0.46 (0.37) | 0.68 (0.53) | 0.67 (0.90) | 0.43 (0.59) |
| EF Babs 870 | 2.03 (1.58) | 2.17 (1.77) | 3.21 (2.51) | 0.51 (4.26) | 2.04 (2.84) |
| EF Babs 401 | 9.01 (2.60) | 14.44 (6.28) | 14.24 (7.55) | 4.03 (0.09) | 7.86 (1.46) |
| EF Babs 401 (BrC) | 5.19 (0.61) | 10.58 (5.07) | 6.37 (2.83) | 3.99 (0.10) | 6.17 (3.74) |
| EF Bscat 870 | 4.51 (2.51) | 9.95 (7.74) | 2.27 (1.12) | 6.72 (7.62) | 22.20 (5.86) |
| EF Bscat 401 | 51.26 (7.87) | 84.03 (55.92) | 35.29 (11.16) | 93.14 (2.45) | 139.45 (153.31) |
| SSA 401 | 0.85 (0.05) | 0.80 (0.05) | 0.70 (0.17) | 0.96 (<0.01) | 0.89 (0.10) |
| SSA 870 | 0.71 (0.08) | 0.71 (0.13) | 0.48 (0.27) | 0.99 (<0.01) | 0.89 (0.15) |
| AAE | 2.26 (0.36) | 2.69 (0.36) | 1.86 (0.20) | 7.13 (0.06) | 4.60 (3.73) |
| MCE | 0.94 (<0.01) | 0.92 (0.0.1) | 0.93 (0.02) | 0.87 (0.62) | 0.86 (0.12) |

[a]Douglas fir, Engelmann spruce, Lodgepole pine, Ponderosa pine, Subalpine fir
[b]Douglas fir, Engelmann spruce, Lodgepole pine, Ponderosa pine, Juniper, Subalpine fir
[c]Douglas fir, Loblolly pine, Lodgepole pine, Ponderosa pine, Subalpine fir
[d]Engelmann spruce, Jeffrey pine, Ponderosa pine, Subalpine fir
10   [e]Douglas fir, Ponderosa pine





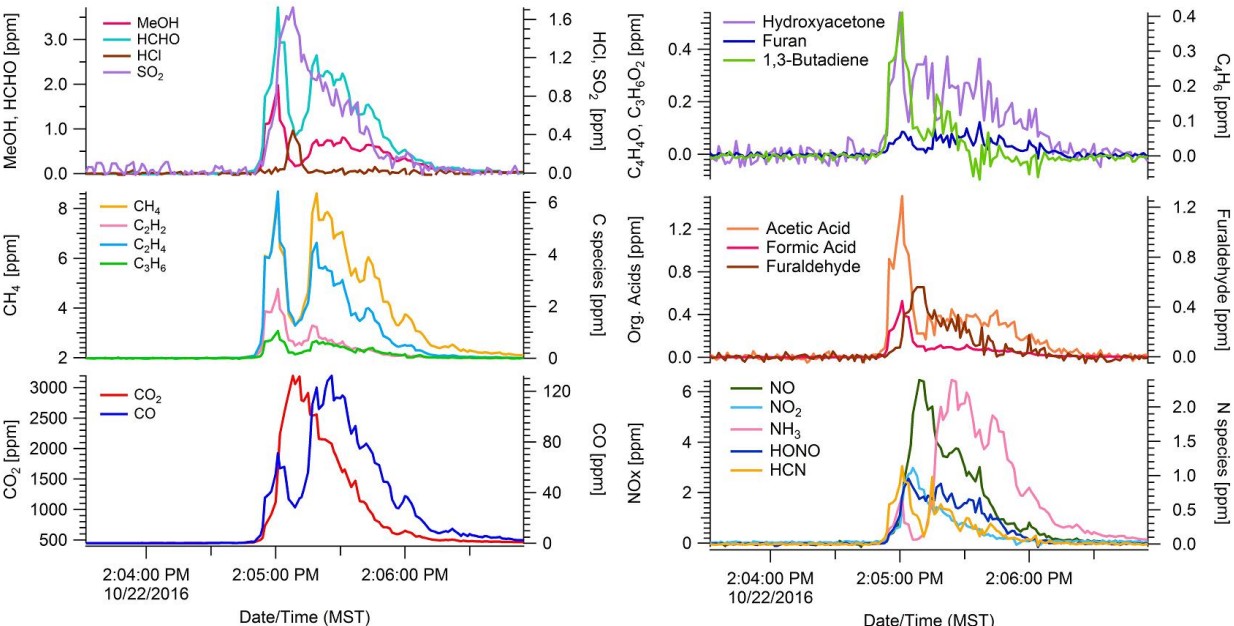

**Figure 1.** Excess mixing ratios of 21 trace gases vs time for a complete Juniper canopy "stack" burn (#75) as measured by OP-FTIR. $CO_2$ denotes flaming, CO denotes smoldering. 1,3-butadiene is shown as an example of lower signal to noise data, but retained since there is no evidence of bias.




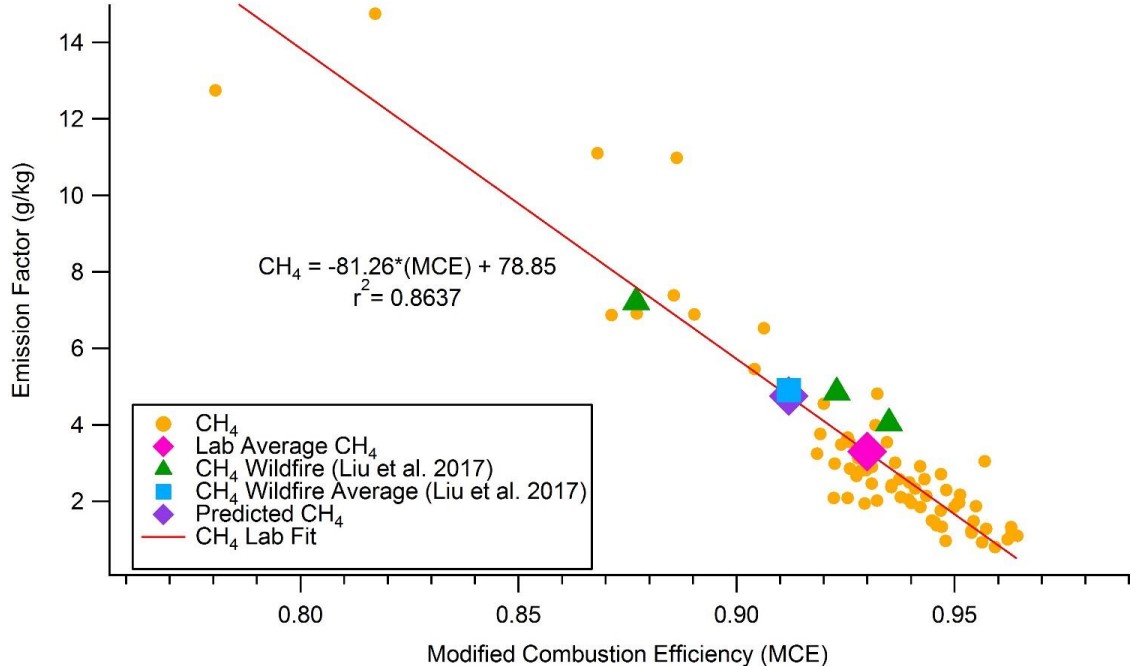

**Figure 2.** Methane emissions from 75 stack fires plotted against corresponding MCE and wildfire field methane emissions plotted against corresponding wildfire field MCE. Also included are the field average methane emissions (blue) and the predicted methane emissions (purple) using the linear regression shown and a field average MCE of 0.912.



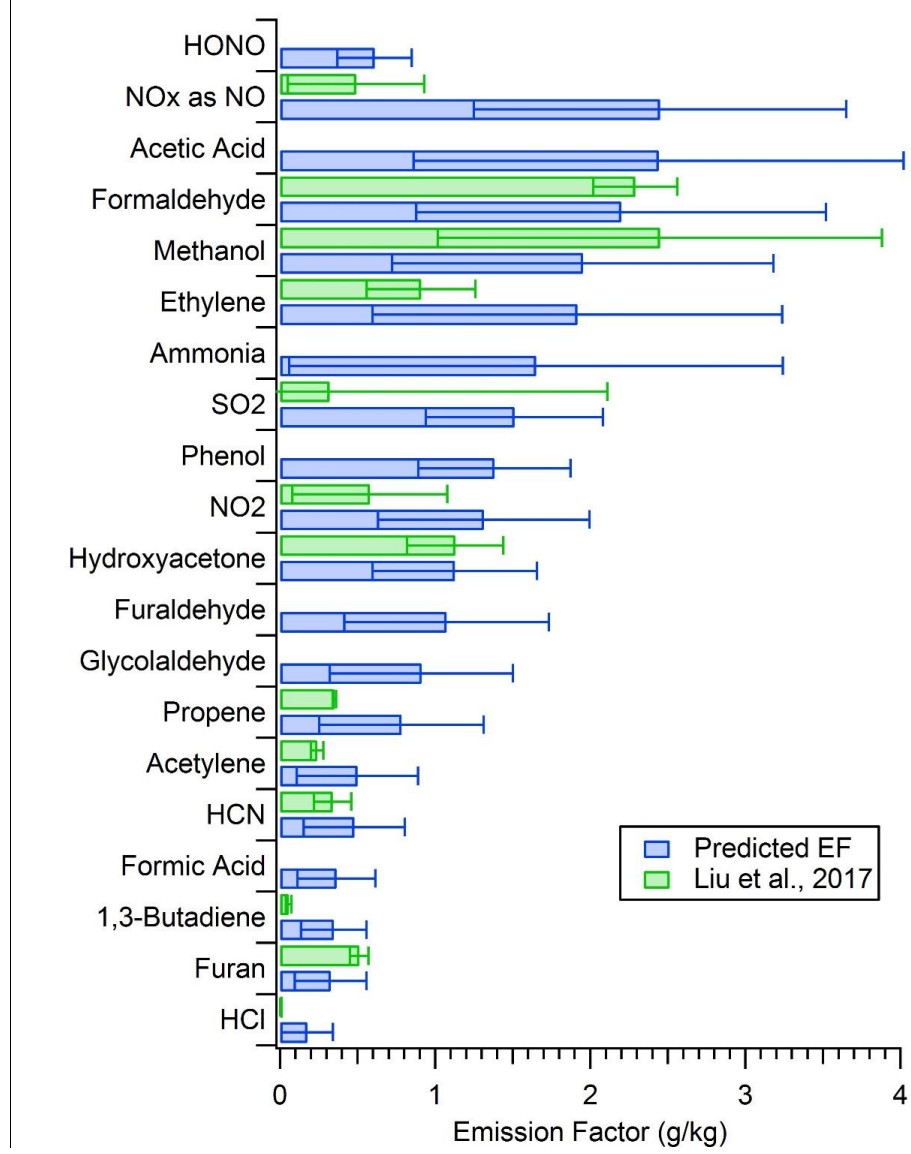

**Figure 3.** Comparison of the lab-predicted EFs at the field average MCE to average field-measured EFs reported by Liu et al. (2017).



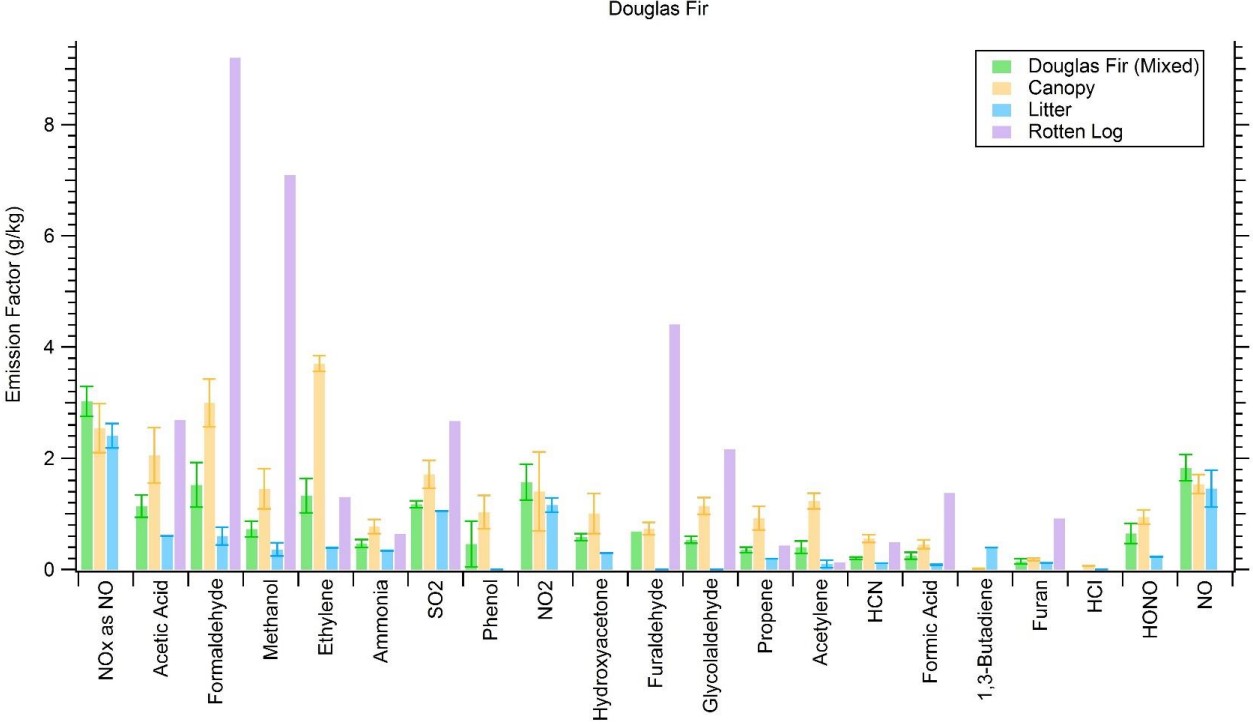

**Figure 4.** Trace gas emissions from mixed Douglas fir ecosystem (including sound, dead wood, but rotten log not included) and pure components. Sound dead wood was not burned separately except as untreated lumber.



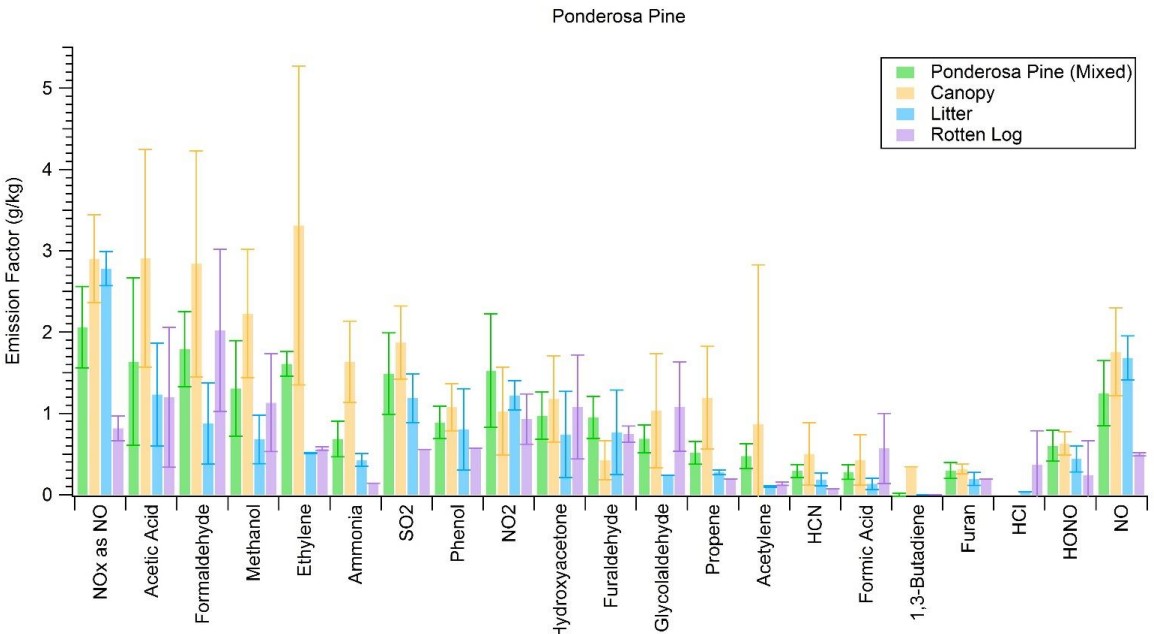

**Figure 5.** Trace gas emissions from mixed Ponderosa pine ecosystem (including sound dead wood, rotten log not included) and pure components.



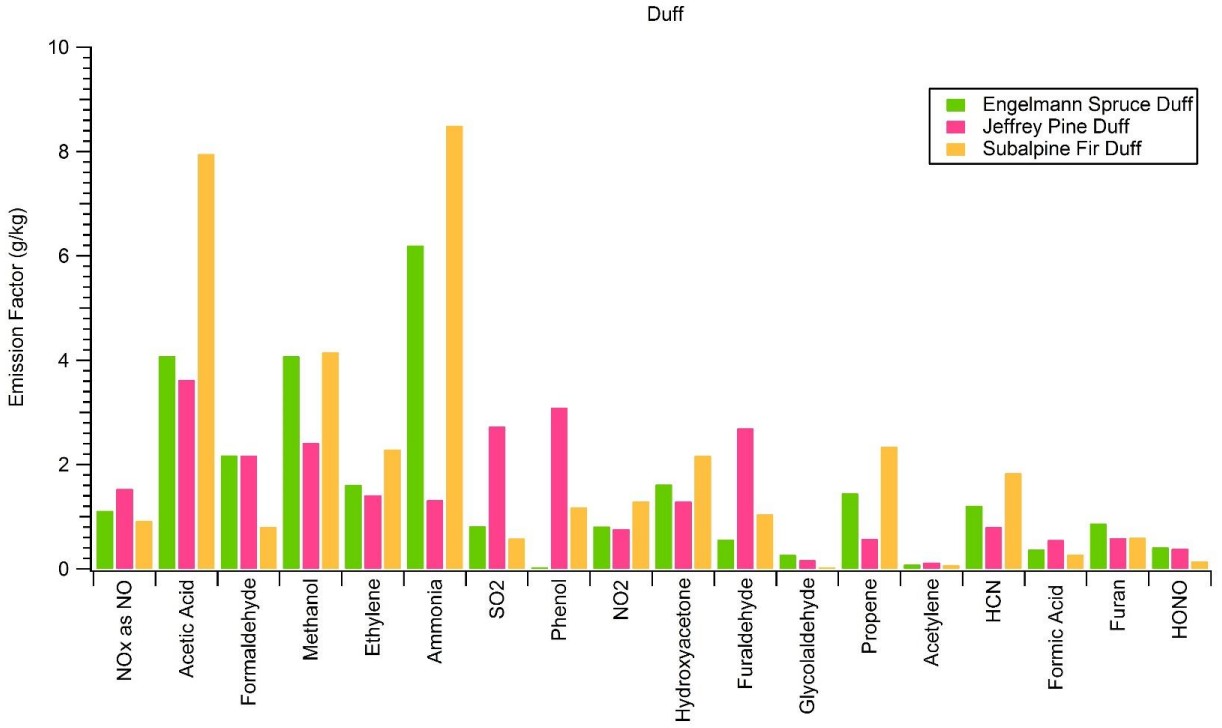

**Figure 6.** Trace gas emissions from pure duff of three different ecosystem types.



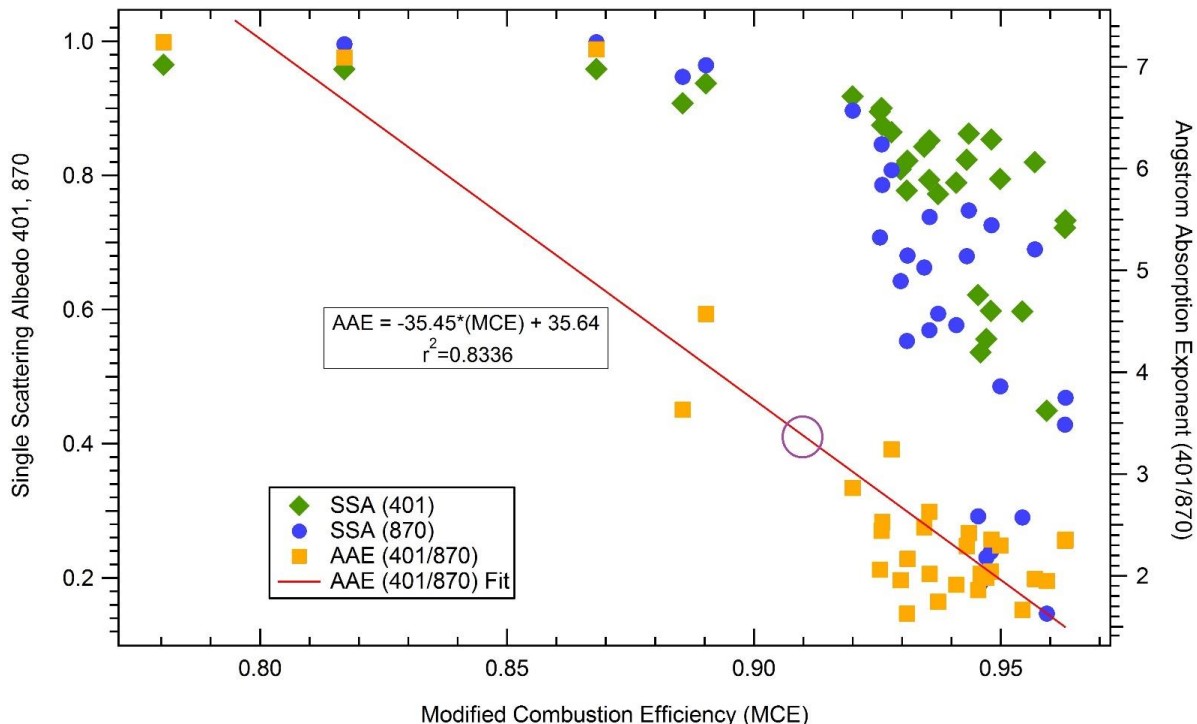

**Figure 7.** SSA at both wavelengths (401, 870 nm) and AAE (401/870) against MCE for 31 stack fires where both 401 and 870 nm data was available. The circle on the fit line represents the lab-predicted AAE using the wildfire field average MCE of 0.912. SSA is difficult to fit to MCE and fits better to EC and OC data, which were not available (Liu et al., 2014; Pokhrel et al., 2016).



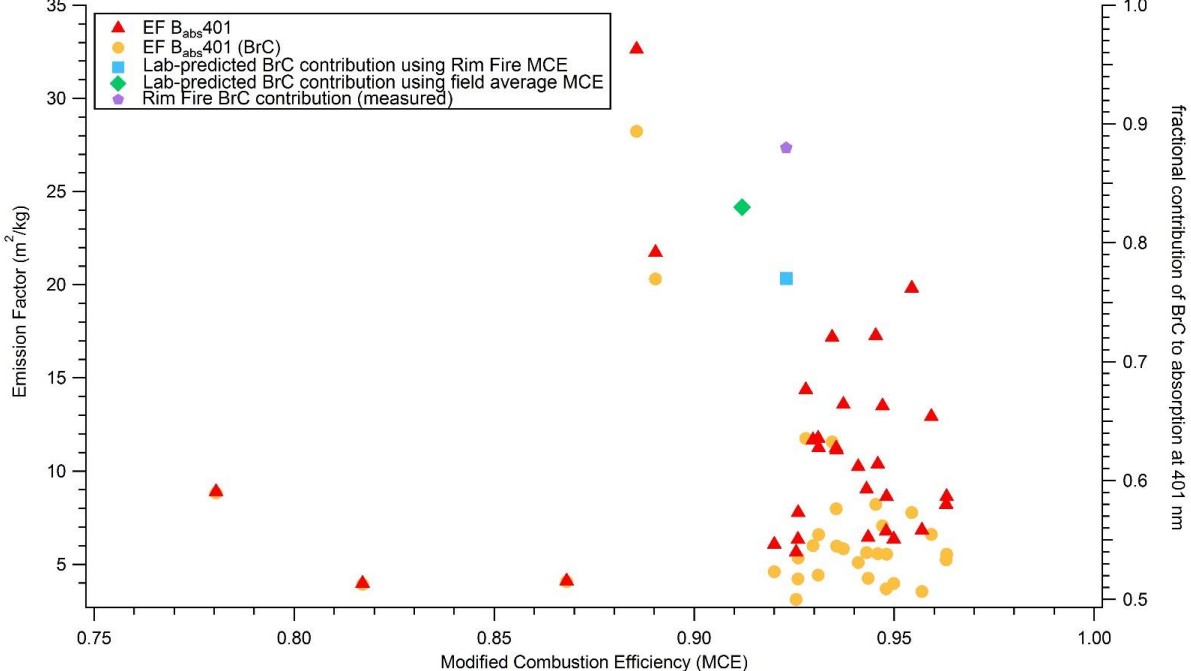

**Figure 8.** Absorption emission factors measured at 401 nm for "BC plus BrC" and for "BrC only" for 31 lab fires, Also shown are the fractional contributions of BrC to total absorption at 401 predicted from the lab AAE data at the field average MCE (green), the Rim Fire MCE (blue) and the field measured AAE (purple) (Forrister et al., 2015; Liu et al., 2017).