# Peer review of "Aerosol optical properties and trace gas emissions by PAX and OP-FTIR for laboratory-simulated western US wildfires during FIREX"

_Atmospheric Chemistry and Physics, 2017_

## Referee Comment (RC1) · Anonymous Referee #1 · 28 Nov 2017

This paper reported emission factors (EFs) of trace gases and optical properties of aerosol particles during combustion of canopy, litter, duff, dead wood in US and some other fuels in the laboratory. The data obtained in this study are valuable to evaluate the impact of biomass burning on climate and atmospheric environment. However, improvement of the discussion will be necessary before considering the publication in ACP.

General comments

[Figure]

Discussion on the relation between the EF for gaseous species and aerosol optical properties is not enough. Especially, optical properties of BrC should depend on chemical compositions of particles and may be indirectly related to the relative EF of gaseous compounds. I recommend adding discussions on this point.

Specific comments

1) Page 5, lines 36-39

Scrubber and diffusion dryer were used in this study. Information on the removal efficiency of light absorbing gases and the particle losses should be added.

2) Page 6, lines 15-24

"The emission ratios to $CO_2$ were then used to derive EFs calculated by the carbon mass balance method (CMB), which assumes all of the burned carbon is volatilized and that all of the major carbon-containing species have been measured" "Our estimate of total carbon in this paper includes these three species and all the rest of the C-containing emissions measured by the OP-FTIR and the PAXs."

These two sentences are confusing. Did the authors include BC in the estimation of total carbon?

3) Page 7, lines 15-19

The authors assume the AAE for BC to be 1 to estimate EF abs405 for BrC. However, this assumption is not necessarily correct (e.g., Lack et al. 2010). In addition, the authors assume that the lensing effect was negligible. However, it is strongly depend on the relative amount of OC to BC, as well as combustion conditions. Is it reasonably to estimate that OC (or BrC) did not coat BC even when OC/BC ratio was very high under low MCE conditions? Discussion on the effect of these assumption (and uncertainties) on the uncertainties in EF abs405 for BrC and EF BC should be added.

4) Page 9, line 20-21

"The lab-measured EFs for these OP-FTIR species and the data for many NMOG species measured by MS and FIREX data in general can thus be used to generate representative EFs or other data for real wildfires."

Because the data of MS was not presented, I recommend avoiding the discussion based on MS data.

5) Page 9, lines 23-32

Because I cannot access to the in preparation papers (Koss et al. and Sekimoto et al.), we (readers) cannot check reasonability of the suggestions.

6) Page 9, lines 43-44

"However, for both vegetation types we observed an enhancement in NOx emissions from the litter and canopy components,.."

Figure 4 shows EF for NOx from litter and canopy were smaller with "Douglas Fir (Mixed)".

7) Page 10, lines 16-18

"As mentioned previously, we measured absorption and scattering coefficients directly at 401 and 870 nm. For the first 31 stack fires, which includes most of the studied fuel types, we have both 401 and 870 nm data. For the remaining 44 stack fires, we only report data at 870 nm as we used our 401 nm PAX for intercomparison studies that will be reported elsewhere."

Same information was given many times (introduction section and experimental section). I recommend avoiding duplicate contents.

8) Page 10, lines 23-24

"Table 3 does not reveal a strong ecosystem dependence among coniferous ecosystems tested for optical properties, but does indicate that chaparral fire aerosol is relatively more absorbing and that there are significant contributions of absorption by BrC at 401 nm among all ecosystems."

The description may be incorrect. Table 3 indicates that EF Babs 870 and EF Babs 401 for chaparral fire aerosols were not necessarily greater than those for Lodgepole Pine and Ponderosa Pine aerosols.

9) Table 4

Because lab. average EF for BC would be calculated from average EF Babs by multiplying a constant factor, I think that the relative uncertainties for them should be same. Why are the ratios 0.53/0.58 and 3.20/5.16 different?

Why did the authors choose different types of function (linear and power law) for EF for Babs401 and EF for Babs401(BrC) for the equation to estimate EF from MCE?

Fitting uncertainties (or reasonability) for each equation should also be added. For example, no clear relations between EF Babs 401 and MCE and between EF Babs401(BrC) and MCE are observed in Fig. 8.

10) Figure 8

I recommend adding error bars.

11) Same terms should be used throughout the text.

For example, EFabs 401, EFabs401, and EF Babs 401 are used for EF for Babs at 401 nm.

References

Lack, D. A. and Cappa, C. D.: Impact of brown and clear carbon on light absorption enhancement, single scatter albedo and absorption wavelength dependence of black carbon, Atmos. Chem. Phys., 10, 4207–4220, doi:10.5194/acp-10-4207-2010, 2010.

---

## Referee Comment (RC2) · Anonymous Referee #2 · 4 Dec 2017

This is a very important and generally well-written manuscript reporting on characterization of gaseous and particulate emission from the laboratory burning of a multitude of Wildland fuels. However, comparisons to results from previous laboratory studies, especially for aerosol emissions and their optical properties are largely missing and errors are not quantified in many figures. This manuscript is appropriate for ACO and should be published after these shortcoming have been corrected and the comments below have been taken into account.

1. P2,L33, 37: Replace the technobabble "lab" with "laboratory" here and elsewhere.

[Figure]

2. Introduction: The work presented here needs to be put into the context of the earlier laboratory studies of aerosol emissions and optical properties including the FLAME study, also conducted at the FSL in Missoula, MT; references to earlier laboratory studies and comparison of results are completely missing. For example, the fact that emissions from the combustion of duffs have a very high AAE (P11, L32) has been reported from a previous FLAME study (Chakrabarty et al., 2010). References and comparisons of emissions from peat and rice straw combustion are also missing.

3. P5,L24-42: References for the PAX instrument including reciprocal nephelometer are mostly missing.

4. P7,L29: Replace "The EFs for scattering and absorption..." with "The EFs for scattering and absorption cross-sections..." to better define what you are actually reporting.

5. P8,L30-31: " It is important to compare our FIREX lab fire emissions data to field measurements of real wildfires to assess how representative and useful the lab-based data are, especially for the many species measured in the lab, but not the field." This seems pretty nonsensical, how do you compare laboratory data with field data for species that weren't measured in the field. Please explain!

6. P8, L41-44: "...because the lab fires had higher average MCE (i.e. a higher fire-integrated flaming/smoldering ratio) than the real wildfires sampled to date, most likely due to some unavoidable drying of the fuels during storage." The second reason may be that in the laboratory, one burns fairly small pieces of fuel, while in the field larger pieces (e.g., tree trunks) may smolder for days.

7. P8, L43 & P9, L30: Please define the "flaming/smoldering ratio"!

8. Error bars must be added to figs. 2, 6, 7, and 8.

REFERENCES Chakrabarty, R. K., H. Moosmuller, L.-W. A. Chen, K. Lewis, W. P. Arnott, C. Mazzoleni, M. Dubey, C. E. Wold, W. M. Hao, and S. M. Kreidenweis (2010).

Brown Carbon in Tar Balls from Smoldering Biomass Combustion. Atmos. Chem. Phys., 10(13), 6363-6370.

---

## Author Comment (AC1) · 8 Jan 2018

Response to Referee #1

We thank the Referee for their interest in our work and the timely and useful comments, which have improved the paper. In the text below we reproduce each comment followed by our response and an exact description of any changes in the revised paper.

Anonymous Referee #1, This paper reported emission factors (EFs) of trace gases and optical properties of aerosol particles during combustion of canopy, litter, duff, dead wood in US and some other fuels in the laboratory. The data obtained in this study are valuable to evaluate the impact of biomass burning on climate and atmospheric environment. However, improvement of the discussion will be necessary before considering the publication in ACP.

General comments

Discussion on the relation between the EF for gaseous species and aerosol optical properties is not enough. Especially, optical properties of BrC should depend on chemical compositions of particles and may be indirectly related to the relative EF of gaseous compounds. I recommend adding discussions on this point.

We agree that it is well-established that certain gases and BC are associated with flaming, while smoldering is associated with BrC and other gases. We now added a description of these indirect associations for completeness on page 10 lines 19-21 as described in detail below.

We are not yet aware of direct mechanistic links between the gaseous species and the aerosol optical properties measured in these fires. We agree aerosol optical properties are related to particle chemistry, but our group did not collect particle chemistry data aligned with the PAX data reported here. Any relevant particle chemistry that was measured by other groups still needs to be analyzed and published. Happily, in the room burn phase of FIREX, our PAXs and extensive other instruments for chemistry and other properties of particles were co-deployed. We think that work will do a better job of addressing the Referee's suggestion than we could do with the data available now. We've provided cross-references to upcoming papers led by other groups already on P3, L8-9 (e.g. Wagner et al, Li et al, both in preparation).

P10, L19-21:

Existing text: "High AAE is an indicator of BrC and relates to smoldering, which is denoted by low MCE and high SSA values. Low AAE, along with low SSA and high MCE values, indicates more flaming combustion."

New text: "High AAE is an indicator of BrC and relates to smoldering, which is denoted by low MCE and high SSA values. Smoldering is also associated with higher EFs for OA, most NMOG, and other gases such as NH3. Low AAE, along with low SSA and high MCE values, indicates more flaming combustion, which is also generally associated with higher EF for BC and "flaming compounds" such as $CO_2$, NOx, and $SO_2$."

Specific comments

1) Page 5, lines 36-39: Scrubber and diffusion dryer were used in this study. Information on the removal efficiency of light absorbing gases and the particle losses should be added.

The manufacturer specification on the scrubber is "minimum removal efficiency of 99.5%" (https://www.purafil.com/wp-content/uploads/2014/12/Purafil-SP-Media-Bulletin.pdf). If the scrubber is not effective this can be detected as absorption even when filtering particles during zeros. However, the scrubber color changes from purple to mostly brown before its effectiveness drops. The scrubber capacity 32 g NO2 per 100 g scrubber combined with the amount we used (~700g) should be sufficient for hundreds of fires, but we exchanged the scrubber pre-emptively partway through the experiment and before any signs of deterioration were observed. We confirm that our drier was a diffusion drier, which we now specify. We did not measure particle losses in the diffusion dryer, but several on-line descriptions claim that "particle loss is minimal in diffusion dryers because the aerosol doesn't contact the desiccant" (e.g. http://dropletmeasurement.com/dmt-diffusion-dryer http://www.tsi.com/diffusion-dryer-3062/ https://www.topas-gmbh.de/en/produkte/ddu-570/). A similar diffusion design is used in the scrubber.

The existing text was: "From the splitter, each separate sample line encountered a scrubber (Purafil-SP Media) to remove absorbing gases and then a drier (Silica Gel 4-10 mesh) to remove water, with this order ensuring that both instruments were sampling at the same relative humidity (varying between 13 and 30%)."

The new text is: "From the splitter, each separate sample line encountered a scrubber to remove absorbing gases (Purafil-SP Media, minimum removal efficiency 99.5%) and then a diffusion drier (Silica Gel 4-10 mesh) to remove water, with this order ensuring that both instruments were sampling at the same relative humidity (varying between 13 and 30%). The scrubber and drier were refreshed before any signs of deterioration were observed (e.g. color change) and the diffusion-based designs should incur minimal particle losses, but losses were not explicitly measured."

2) Page 6, lines 15-24: "The emission ratios to CO2 were then used to derive EFs calculated by the carbon mass balance method (CMB), which assumes all of the burned carbon is volatilized and that all of the major carbon-containing species have been measured" "Our estimate of total carbon in this paper includes these three species and all the rest of the C-containing emissions measured by the OP-FTIR and the PAXs." These two sentences are confusing. Did the authors include BC in the estimation of total carbon?

Yes—BC was measured by PAX and then coupled with OP-FTIR data to compile an emission factor (EF BC), which was included in the CMB along with other carbon-containing species that we measured.

We modified this sentence to read "Our estimate of total carbon in this paper includes these three species and all the rest of the C-containing gases measured by the OP-FTIR as well as the C in the particles (i.e. BC and OC) based on the PAX data."

To further clarify on P7, L24-26 we updated the text to read: "We use the qualitative OA to calculate a small term in our CMB that helps account for unmeasured C-species (assuming OA/OC of 1.6), but we do not report OA or OC in the tables as quantitative species."

3) Page 7, lines 15-19: The authors assume the AAE for BC to be 1 to estimate EF abs405 for BrC. However, this assumption is not necessarily correct (e.g., Lack et al. 2010). In addition, the authors assume that the lensing effect was negligible. However, it is strongly depend on the relative amount of OC to BC, as well as combustion conditions. Is it reasonably to estimate that OC (or BrC) did not coat BC even when OC/BC ratio was very high under low MCE conditions? Discussion on the effect of these assumption (and uncertainties) on the uncertainties in EF abs405 for BrC and EF BC should be added.

It is important that the attribution of BrC versus lensing is more uncertain than the absorption data itself. To allude to this we had provided an uncertainty of +/- 20% in the AAE for BC and acknowledged up to 30% absorption from coatings in older room burn smoke (Pokhrel et al 2017). Additional measurements (in preparation) during the FIREX room burns with a thermodenuder indicated that lensing enhancements were typically 5-10% at 870 nm even in smoke 15 minutes to hours old. Thus, again in the 5 second old smoke for stack burns it's likely the lensing effects are smaller as already noted. Without supporting measurements in the stack we can only make a best estimate of the lensing contribution based on closely-related work.

To improve the depiction of this uncertainty in BrC attribution we have added an uncertainty estimate for the BrC attribution (+/- 25%) to line 21 and we now refer to "OA absorption due mainly to BrC at 401 nm" on line 26. We also added an illustrative error bar to Fig. 8 (as also requested by Referee #2) and added the suggested reference to Lack and Cappa, 2010.

4) Page 9, line 20-21: "The lab-measured EFs for these OP-FTIR species and the data for many NMOG species measured by MS and FIREX data in general can thus be used to generate representative EFs or other data for real wildfires." Because the data of MS was not presented, I recommend avoiding the discussion based on MS data.

We think this is a key finding that is important to retain. The representativeness of the FIREX fires is a general issue of great importance to all the participating groups. Our probe/discussion of that issue is not based on MS or other data, but successfully shows that reasonably realistic values can be extracted from the lab fires based on comparing the diverse lab-field overlap species measured by the OP-FTIR. That then has the important implication that the MS and other data are useful to represent real fires.

To clarify on P8, L31 we added: "We assess representativeness by comparing the EF results for species measured in both the field and our laboratory fires."

5) Page 9, lines 23-32: Because I cannot access to the in preparation papers (Koss et al. and Sekimoto et al.), we (readers) cannot check reasonability of the suggestions.

Koss et al is now available via ACPD and we updated the reference. The point of referring to these papers is not that our approach depends on them, but to make readers aware of other data and approaches that they may find useful. We think it will be useful to retain and update cross-referencing to the other closely related work.

6) Page 9, lines 43-44: "However, for both vegetation types we observed an enhancement in NOx emissions from the litter and canopy components,.." Figure 4 shows EF for NOx from litter and canopy were smaller with "Douglas Fir (Mixed)".

Thank you. We fixed the sentence to say, "Additionally, we observed an enhancement in NOx emissions from the litter and canopy components in Ponderosa Pine." We also updated the sentence in the conclusions P12, L41-43 to read: For instance, emissions of some NMOG were enhanced from a Douglas Fir rotten log and emissions of NOx were enhanced from Ponderosa Pine litter and canopy components.

7) Page 10, lines 16-18: "As mentioned previously, we measured absorption and scattering coefficients directly at 401 and 870 nm. For the first 31 stack fires, which includes most of the studied fuel types, we have both 401 and 870 nm data. For the remaining 44 stack fires, we only report data at 870 nm as we used our 401 nm PAX for intercomparison studies that will be reported elsewhere." Same information was given many times (introduction section and experimental section). I recommend avoiding duplicate contents.

We chose to duplicate these comments because several of our coauthors suggested "refreshers" for the reader. Some readers may skip directly to the optical property section of the paper, so we opted to leave this as is.

8) Page 10, lines 23-24: "Table 3 does not reveal a strong ecosystem dependence among coniferous ecosystems tested for optical properties, but does indicate that chaparral fire aerosol is relatively more absorbing and that there are significant contributions of absorption by BrC at 401 nm among all ecosystems." The description may be incorrect. Table 3 indicates that EF Babs 870 and EF Babs 401 for chaparral fire aerosols were not necessarily greater than those for Lodgepole Pine and Ponderosa Pine aerosols.

The Referee is right as written. We changed "is relatively more absorbing" to "has consistently lower SSA than coniferous fire aerosol" – the new text is correct and should clarify the point we actually intended to make.

9) Table 4: Because lab. average EF for BC would be calculated from average EF Babs by multiplying a constant factor, I think that the relative uncertainties for them should be same. Why are the ratios 0.53/0.58 and 3.20/5.16 different?

The initial EF for BC was incorrect. Rather than being the average for all 75 stack fires where 870 nm data was available, we incorrectly only reported the average of the first 31. We fixed this to now correctly account for all 75 stack fires. It has been corrected to: 0.67 (1.09). The EF Babs 870 number was correct and was not changed.

Why did the authors choose different types of function (linear and power law) for EF for Babs401 and EF for Babs401(BrC) for the equation to estimate EF from MCE?

This was empirical, we just chose whatever fit best based on the $r^2$ value, but confirmed the fit looked reasonable especially near the field-average MCE for wildfires.

Fitting uncertainties (or reasonability) for each equation should also be added. For example, no clear relations between EF Babs 401 and MCE and between EF Babs401(BrC) and MCE are observed in Fig. 8.

We've added the r-squared value for each equation in a new column. Due to a few outliers the EFBabs at 401 is not highly correlated with MCE, but the equation nevertheless estimates reasonable looking values averaged over the scatter for MCE values near the wildfire field average MCE. This is now clarified in a footnote at the head of the r-squared column "The low r2 equations return reasonable values near the field average MCE." In general, the application of these equations is too extract a reasonable value from the data at an important field-measured characteristic value rather than demonstrate high correlation, which may not in fact exist. Stated alternatively, EFBabs401 is likely impacted by many factors and doesn't correlate highly with MCE, nonetheless a reasonable guess at the appropriate EFBabs401 at the field average flaming/smoldering characteristics of wildfire is important and can be obtained from the equations presented.

10) Figure 8: I recommend adding error bars.

We have added representative error bars to Fig. 8 and other figures as requested by Referee #2.

11) Same terms should be used throughout the text. For example, EFabs 401, EFabs401, and EF Babs 401 are used for EF for Babs at 401 nm.

We fixed all terms to now either say EF $B_{abs}$401 or EF $B_{scat}$401. The same goes for the 870 nm values.

References

Lack, D. A. and Cappa, C. D.: Impact of brown and clear carbon on light absorption enhancement, single scatter albedo and absorption wavelength dependence of black carbon, Atmos. Chem. Phys., 10, 4207–4220, doi:10.5194/acp-10-4207-2010, 2010.

We've added this reference.

---

## Author Comment (AC2) · 8 Jan 2018

Response to Referee #2

We thank the Referee for their interest in our work and the timely and useful comments, which have improved the paper. In the text below we reproduce each comment followed by our response and an exact description of any changes in the revised paper. (At the end of this response we append a short list of minor voluntary corrections/updates that don't affect any conclusions.)

Anonymous Referee #2, This is a very important and generally well-written manuscript reporting on characterization of gaseous and particulate emission from the laboratory burning of a multitude of Wildland fuels. However, comparisons to results from previous laboratory studies, especially for aerosol emissions and their optical properties are largely missing and errors are not quantified in many figures. This manuscript is appropriate for ACO and should be published after these shortcoming have been corrected and the comments below have been taken into account.

1. P2,L33, 37: Replace the technobabble "lab" with "laboratory" here and elsewhere.

We replaced "lab" with "laboratory everywhere except for in a few hyphenated usages to prevent clumsy long words.

2. Introduction: The work presented here needs to be put into the context of the earlier laboratory studies of aerosol emissions and optical properties including the FLAME study, also conducted at the FSL in Missoula, MT; references to earlier laboratory studies and comparison of results are completely missing. For example, the fact that emissions from the combustion of duffs have a very high AAE (P11, L32) has been reported from a previous FLAME study (Chakrabarty et al., 2010). References and comparisons of emissions from peat and rice straw combustion are also missing.

There have been hundreds of papers describing previous laboratory BB studies at the FSL, Max Planck Institute, India, and elsewhere dating back to at least 1991 and we've added text to the introduction on P2, L37 before "However":

"For these reasons, numerous laboratory studies have been crucial to advance our understanding of biomass burning emissions (e.g. Lobert et al., 1991; Yokelson et al., 1996; Lewis et al., 2008; McMeeking et al., 2009; etc)."

References added:

Lobert, J. M., D. H. Scharffe, W. M. Hao, T. A. Kuhlbusch, R. Seuwen, P. Warneck, and P. J. Crutzen.: Experimental evaluation of biomass burning emissions: Nitrogen and carbon containing compounds, in Global Biomass Burning: Atmospheric, Climatic, and Biospheric Implications, edited by J. S. Levine, MIT Press, Cambridge, Mass., 1991.

McMeeking, G. R., Kreidenweis, S. M., Baker, S., Carrico, C. M., Chow, J. C., Collet Jr., J. L., Hao, W. M., Holden, A. S., Kirchstetter, T. W., Malm, W. C., Moosmüller, H., Sullivan, A. P., and Wold, C. E.: Emissions of trace gases and aerosols during the open combustion of biomass in the laboratory, J. Geophys. Res., 114, D19210, doi:10.1029/2009JD011836, 2009.

However, to our knowledge this study is the first to focus specifically on simulation of wildfires. Thus, we agree it makes sense to add a comparison to the FLAME duff-combustion results the Referee recommends since it is an overlapping fuel with our study. At P11, L44 we have added the following:

"We can compare our duff results to previous measurements of optical properties of duff-fire aerosol by Chakrabarty et al (2010). These authors identified tarballs as a major BrC species produced by duff combustion and they measured an AAE of 4.2 (405 and 532 nm wavelength pair) for a Ponderosa Pine duff sample from MT. Including their other duff sample (AK feather moss duff), they obtained a study-average duff-combustion AAE of 5.3. We measured AAE on two much larger burns (~4 times more fuel mass, Fires # 12 and 26) in Engelmann Spruce duff, with different wavelengths, and at much lower MCE ($0.843 \pm 0.036$ versus ~0.91). We obtained a study-average duff combustion AAE of 7.13 (0.057). Both studies observed a high AAE for duff combustion. Their lower AAE values could be related to different wavelengths used, the possibility of some BrC abs at 532 nm (Bluvshtein et al., 2017), the different duff type, and/or their higher MCE, which they attributed to sampling some flaming combustion during the ignition process. The AAE calculated from our AAE versus MCE fit (for all fuels) at their MCE of 0.91 is relatively closer to their value."

New references:

Bluvshtein, N., P. Lin, J. M. Flores, L. Segev, Y. Minon, E. Tas, G. Snyder, C. Weagle, S. S. Brown, A. Laskin, and Y. Rudich, Broadband optical properties of biomass-burning aerosol and identification of brown carbon chromophores, J. Geophys. Res., 122, doi:10.1002/2016JD026230, 2017.

Chakrabarty, R. K., H. Moosmuller, L.-W. A. Chen, K. Lewis, W. P. Arnott, C. Mazzoleni, M. Dubey, C. E. Wold, W. M. Hao, and S. M. Kreidenweis.: Brown carbon in tar balls from smoldering biomass combustion, Atmos. Chem. Phys., 10(13), 6363-6370, 2010.

Peat and rice straw (and dung) are very minor components of this study used only briefly to check fuel chemistry effects or compare to field data to further investigate the possibility of reasonably realistic simulations in the laboratory. In addition, an exhaustive discussion of these fuels could potentially include some previous lab studies that may have had less realistic results. For instance some previous lab studies of peat fire emissions reported unrealistic EC emissions by the thermal method or C2H2/CH4 ratios >1 where the latter shows that the emissions sampled were actually dominated by the propane torch used for ignition. We prefer not to engage in a lengthy discussion of these issues in this paper about wildfires. Finally, there are recently

published, more extensive, lab and field comparisons for peat and rice straw combustion, which are noted in our new text revised as follows:

P12, L13: Cited Pokhrel et al., 2016 peat AAE paper after first "AAE"

P12, L26: We added "briefly" before "summarized"

P12, L28: We appended to the end of the paragraph: "More comprehensive, recent discussions of these fuels can be found elsewhere (Stockwell et al., 2016a, b; Jayarathne et al., 2017a, b)."

References:

Jayarathne, T., Stockwell, C. E., Bhave, P. V., Praveen, P. S., Rathnayake, C. M., Islam, Md. R., Panday, A. K., Adhikari, S., Maharjan, R., Goetz, J. D., DeCarlo, P. F., Saikawa, E., Yokelson, R. J., and Stone, E. A.: Nepal Ambient Monitoring and Source Testing Experiment (NAMaSTE): Emissions of particulate matter from wood and dung cooking fires, garbage and crop residue burning, brick kilns, and other sources, Atmos. Chem. Phys. Discuss., https://doi.org/10.5194/acp-2017-510, in review, 2017a.

Jayarathne, T., Stockwell, C. E., Gilbert, A. A., Daugherty, K., Cochrane, M. A., Ryan, K. C., Putra, E. I., Saharjo, B. H., Nurhayati, A. D., Albar, I., Yokelson, R. J., and Stone, E. A.: Chemical characterization of fine particulate matter emitted by peat fires in Central Kalimantan, Indonesia, during the 2015 El Niño, Atmos. Chem. Phys. Discuss., https://doi.org/10.5194/acp-2017-608, in review, 2017b.

3. P5,L24-42: References for the PAX instrument including reciprocal nephelometer are mostly missing.

P5, L25: We added a reference to an earlier prototype instrument with some similarities to our PAXs in that they combined a reciprocal neph with a PAS (Lewis et al., 2008). We had already cited a recent detailed "PAX description and evaluation" paper by Nakayama et al., 2015

Reference:

Lewis, K., Arnott, W. P., Moosmuller, H., and Wold, C. E.: Strong spectral variation of biomass smoke light absorption and single scattering albedo observed with a novel dual-wavelength photoacoustic instrument, J. Geophys. Res., 113, D16203, doi:10.1029/2007JD009699, 2008.

4. P7, L29: Replace "The EFs for scattering and absorption: : :" with "The EFs for scattering and absorption cross-sections: : :" to better define what you are actually reporting.

We updated the text here to read: "The EFs for scattering and absorption optical cross-sections…".

5. P8,L30-31: " It is important to compare our FIREX lab fire emissions data to field measurements of real wildfires to assess how representative and useful the lab-based data are, especially for the many species measured in the lab, but not the field." This seems pretty nonsensical, how do you compare laboratory data with field data for species that weren't measured in the field. Please explain!

As noted in our response to comment 4 of Referee #1: To clarify on P8, L31 we added: "We assess representativeness by comparing the EF results for species measured in both the field and our laboratory fires."

6. P8, L41-44: ": : :because the lab fires had higher average MCE (i.e. a higher fire-integrated flaming/smoldering ratio) than the real wildfires sampled to date, most likely due to some unavoidable drying of the fuels during storage." The second reason may be that in the laboratory, one burns fairly small pieces of fuel, while in the field larger pieces (e.g., tree trunks) may smolder for days.

The largest diameter dead/down woody debris fuels are referred to as 1000 hr fuels and are over 7.6 cm in diameter (Table S1). We burned some of these fuels, but upon re-checking we do find that they were under-represented by our team of forest fire "experts" compared to the FOFEM-recommended amounts. We thank the Referee for bringing this to our attention and have appended the following to the end of the sentence: "and some under-representation of the largest diameter fuels (Tab S1)."

P11, L11: deleted "all the" so the sentence doesn't imply "perfection."

In addition, in the conclusions P13, L13: We changed: "Using a simple procedure to account for the flaming to smoldering ratio, we generated EF values from the lab data that were in agreement with the field data for …."    to    "Despite some underrepresentation of the largest diameter fuel class we were able to use a simple procedure to account for the flaming to smoldering ratio and generate EF values from the laboratory data that were in agreement with the field data for …"

7. P8, L43 & P9, L30: Please define the "flaming/smoldering ratio"!

This is explained in different words on page 6 associated with the description of MCE. To clarify, on P6, L36, we appended to the end of the sentence "and an MCE of 0.9 would indicate roughly equal amounts of flaming and smoldering (i.e. a flaming/smoldering ratio of ~1)"

8. Error bars must be added to figs. 2, 6, 7, and 8.

We added representative error bars to each of these figures.

REFERENCES Chakrabarty, R. K., H. Moosmuller, L.-W. A. Chen, K. Lewis, W. P. Arnott, C. Mazzoleni, M. Dubey, C. E. Wold, W. M. Hao, and S. M. Kreidenweis (2010). Brown Carbon in Tar Balls from Smoldering Biomass Combustion. Atmos. Chem. Phys., 10(13), 6363-6370.

This was added as noted above.

We've also made some minor voluntary corrections and updates as described next:

P1, L17: "Subalpine Fire" changed to "Subalpine Fir".

P1, L28: After a last-minute addition of a comparison to the one previous wildfire airborne NH3 measurement, we forgot to update the abstract and conclusion.

The existing text was: "This is especially valuable for species not yet measured in the field. For instance, the OP-FTIR data alone show that ammonia (1.65 g kg-1), acetic acid (2.44 g kg-1), nitrous acid (HONO, 0.61 g kg-1) and other trace gases such as glycolaldehyde and formic acid are significant emissions not previously measured for US wildfires."

This now reads: "This is especially valuable for species rarely or not yet measured in the field. For instance, the OP-FTIR data alone show that ammonia (1.65 g kg-1), acetic acid (2.44 g kg-1), nitrous acid (HONO, 0.61 g kg-1) and other trace gases such as glycolaldehyde and formic acid are significant emissions previously poorly, or uncharacterized, for US wildfires."

P1, L35: removed unmatched ")" after "kg-1".

P4, L12: After "poplar shavings" added "(aka "excelsior")" to connect to name in supplemental tables.

P6, L14 and also on P12, L37: We've added two new gases (C2H2 and C2H4) to the list that we analyzed for in the room burns due to a recent (post-submission) request.

P6, L31: After smoldering we added, "where "smoldering" is an approximate term for all non-flaming processes (e.g. glowing and pyrolysis) as explored in more detail elsewhere (Yokelson et al., 1996, Koss et al., 2017; Sekimoto et al., in preparation)"

P8, L33: "Compositions" corrected to "Composition" (not plural) in SEAC4RS.

P9, L43: Appended "because of a transition to flaming combustion during the second half of the fire."

P10, L25: Corrected quoted lab average AAE from "2.19 ± 0.24" to "2.80 ± 1.57" consistent with conclusions and Table 4.

P10, L36: Liu et al reference, we added year

P11, L14: We removed an unnecessary sentence about Tables 2 and 3.

P11, L15: Added citation to reflect that the Rim Fire AAE was from Forrister et al., 2015

P12, L9 sect 3.7 header: changed to "Trace gas …" (no plural)

P12, L13: added "emissions" after "BC".

P12, L14: corrected EFCH4 from "10.83" to "10.39".

P12, L17: We expanded "(BC extremely small and gases within 31%)" to "(EF BC extremely small compared to most biomass burning (Akagi et al., 2011) and gases within 31%)" to clarify "small"

P12, L22: Added "EF" before "BC" to clarify as above.

P13, L7: changed the "with BrC accounting for nearly 100% and 78% of the absorption at 401, respectively, for these fuel components." to "with BrC accounting for nearly 100% and 94% of the absorption at 401, respectively, for these fuel components (using data only from fires with measurements at two wavelengths)."

P13, L17: The NH3 uniqueness retracted by adding "rarely, or" before "not previously" for reasons explained above.

Table 4: We removed un-needed "EF" from the "Lab AVG EF" column header.

Table 5: The MCE variability for duff was missing a zero and has been fixed.

---

## Author Response (AR2)

**Response to co-editor comments:**

We thank the co-editor for a timely, thorough re-reading of the paper and the good suggestions/corrections, which have improved the paper. Below we reproduce the co-editors exact comments (C#) and then describe exactly how we changed the paper (R#).

Co-Editor Decision: Publish subject to minor revisions (review by editor) (14 Jan 2018) by Sergey A. Nizkorodov

Comments to the Author:

C1. There are too many papers cited as "in preparation", such as Li et al. on page 3, Wagner et al. on page 3, Roberts at al. on page 4, Santin et al. on page 6, Sekimoto et al. on page 6 and 9. I would strongly recommend removing such references to unpublished papers from the text and rewording the sentences containing theses references accordingly. The practice of citing unpublished work is strongly discouraged by the scientific publishers.

R1. The in preparation references were not needed to support our conclusions, but just intended to provide useful connections to other relevant work in a highly collaborative experiment. However, we see the co-editors point and after co-author discussion, we have removed the references and adjusted the text as shown in "track-changes."

C2. Most writers use a comma after "e.g." when it is followed by multiple examples. I just confirmed this by reading technical writing books. You may want to inset these commas.

R2. Thanks. We searched on "e.g." and added a comma in about a half-dozen places.

C3. P1, L17: period missing after "Manzanita"

R3. Thanks, fixed.

C4. P1, L26: should "EF" be changed to "EFs"?

R4. Thanks, fixed.

C5. P1, L30: please reword "are significant emissions previously poorly, or uncharacterized,"

R5. We changed this to: "…trace gases such as glycolaldehyde and formic acid are significant emissions that were poorly characterized or uncharacterized for US wildfires in previous work." We also added the EF in parentheses after these compounds to be consistent with previous text (see track changes)

C6. P4, L30: period missing after et al.

R6. This reference was "in preparation" and therefore removed as requested.

C7. P5, L13: "are based" should be "were based"

R7. Thanks, fixed.

C8. P5, L14: semicolon needed after the Griffith reference

R8. Thanks, fixed.

C9. P5, L35: the readers will not know what is meant by "absorbing gases". I would be more specific here.

R9. We changed to "UV-absorbing gases such as $NO_2$"

C10. P6, L6: "and emission factors" -> ", emission factors"

R10. Thanks. We added the necessary commas to the Sect 2.4 header.

C11. P6, L13: "ER-to-CO2 ratios" should be reworded (or fully spelled out as you do on line 16) because ER already contains ratio in it

R11. It now says "molar ERs to $CO_2$ were calculated"

C12. Equation 1: in my opinion, the use of delta C instead of delta X in the denominator may be confusing because the readers may mistakenly assume that delta C in the denominator is the excess concentration of carbon in a given species (which would make the equation incorrect). But I see that this equation has propagated through multiple uses in previous publications so it is probably OK.

R12. We agree this could be confusing at a glance, but we have defined the parameters in the text and think a careful reader will get our intent correctly. We also agree that maintaining continuity is of value so have not changed anything here.

C13. P7, L32: I think it might be worth explaining the usefulness of EF Babs. (It might have been done in previous papers.) My understanding if that that product of EF Babs (in m2/kg) and amount of fuel burned normalized by the volume of air in which the smoke diluted into (in kg/m3) would give you the absorption coefficient for the air containing the smoke. Would it be worth stating this or providing a formula? This can also be done in section 3.4.

R13. Great suggestion. We added: ""$EF_{abs}$ or $EF_{scat}$ are more precisely the optical cross-section (in $m^2$) due to the particles produced when a kg of fuel is burned if the emissions are mixed into a $m^3$ of air. These EFs enable direct calculation of the absorption or scattering coefficient (in $m^{-1}$) by multiplication with a specified ratio of fuel burned to a volume of air (kg $m^{-3}$) (Bond et al., 1999; Moosmüller et al., 2005)."

We realized we also had two shorthand notations so we also adopted the shorter and less confusing notation throughout by changing from (e.g.) "EF $B_{abs}$ 401" to "$EF_{abs}$401", with no spaces in the text, figure 8 legend, and tables.

References added:

Bond, T. C., Bussemer, M., Wehner, B., Keller, S., Charlson, R. J., and Heintzenberg, J.: Light absorption by primary particle emissions from a lignite burning plant, Environ. Sci. Technol., 33, 3887-3891, 1999.

Moosmüller, H., Varma, R., Arnott, W. P., Kuhns, H., Etyemezian, V., and Gillies, J. A.: Scattering cross section emission factors for visibility and radiative transfer applications: Military vehicles traveling on unpaved roads, J. Air & Waste Manage. Assoc., 55, 1743-1750, 2005.

In the course of working on this we discovered a tiny calculation error that increased EF values for the first 31 fires by typically 0.3% and always less than 1%, which is technically insignificant. However, after some deliberation, we updated the entire paper (figures, tables, and text) as needed with the "new" EF to ensure the text description was exact even though there was no impact on conclusions and the changes were very small compared to quoted uncertainties. This is shown in track changes (e.g. see examples in abstract).

C14. P7, L33: "Babs" should contain a subscript

R14. Fixed, thanks.

C15. P10, L8: remove extra period

R15. Thanks. By searching for ".." we found three incorrect extra periods that were removed.

C16. P10, L26: remove extra period

R16. Done.

C17. P10, L27 and P13, L26: "U.S" should be "U.S."

R17. Thanks, fixed both and seven more.

C18. P11, L12 and P13, L10: should there be a space in "EFabs401" to make it consistent with the tables?

R18. Thanks, as noted above, instead we deleted the unneeded spaces and the "B" in EF Babs throughout the paper.

C19. P12, L7: period needed after "et al"

R19. Thanks by searching on "et al_" we found four missing periods that were added.

C20. Table 1: there are too many significant digits specified for some of the values, such as EFs $CO_2$, CO. The numbers are normally truncated such that the standard deviation has only 1 or 2 significant digit. What is the justification for using 1682.95+/-25.03 instead of 1683+/-25?

R20. We do sometimes carry extra digits to eliminate "round-off error" and make the formatting more attractive trusting that users will consider the stated uncertainties. However, we have implemented the suggested change.

C21. Table 1: chemical formulas in the first column should be formatted to include subscripts

R21. Thanks we had noticed this and planned to fix it in proofs, but implemented it now instead.

C22. Table 2: I have a similar comment on significant digits here. The slopes and intercepts have up to 5 significant digits, and this sort of precision is hard to believe when looking at the data plotted in Figure 2. Are the slope and intercept columns even needed in this table? If you keep them, please explain their meaning in the footnote.

R22. In this specific application the formula often calculates a number that is a relatively small difference between two large numbers especially for the smaller NMOG EFs that can occur at some high MCE values. Thus, carrying a number of digits past the decimal point ensures that rounding off doesn't lead to inaccurate results. We have truncated some entries for species with larger EF though. Overall, by including the slope and intercept column, readers can check our calculation or calculate EF at other MCE if needed. We added footnote "a" to explain this and changed the existing footnote "a" to footnote b.

C23. Table 4: Please explain in the footnote what is x and what is y in the equations

R23. Updated footnotes to explain that X is MCE and y is the observable in column 1.

[revised manuscript text omitted]